# How do Variational Autoencoders Learn? Insights from Representational Similarity

## Abstract

The ability of Variational Autoencoders (VAEs) to learn disentangled representations has made them popular for practical applications. However, their behaviour is not yet fully understood. For example, the questions of when they can provide disentangled representations, or suffer from posterior collapse are still areas of active research. Despite this, there are no layerwise comparisons of the representations learned by VAEs, which would further our understanding of these models. In this paper, we thus look into the internal behaviour of VAEs using representational similarity techniques. Specifically, using the CKA and Procrustes similarities, we found that the encoders' representations are learned long before the decoders', and this behaviour is independent of hyperparameters, learning objectives, and datasets. Moreover, the encoders' representations in all but the mean and variance layers are similar across hyperparameters and learning objectives.

## 1 Introduction

Variational Autoencoders (VAEs) are considered state-of-the-art techniques to learn unsupervised disentangled representations, and have been shown to be beneficial for fairness (Locatello et al., 2019a). As a result, VAEs producing disentangled representations have been extensively studied in the last few years (Locatello et al., 2019b; Mathieu et al., 2019; Rolinek et al., 2019; Zietlow et al., 2021), but they still suffer from poorly understood issues such as posterior collapse (Dai et al., 2020). While some work using explainability techniques has been done to shed light on the behaviour of VAEs (Liu et al., 2020), a comparison of the representations learned by different methods is still lacking (Zietlow et al., 2021). Moreover, the layer-by-layer similarity of the representations within models has yet to be investigated.

Fortunately, the domain of deep representational similarity is an active area of research and metrics such as SVCCA (Raghu et al., 2017; Morcos et al., 2018), Procrustes distance (Schönemann, 1966), or Centred Kernel Alignment (CKA) (Kornblith et al., 2019) have proven very useful in analysing the learning dynamics of various models (Wang et al., 2019; Kudugunta et al., 2019; Raghu et al., 2019; Neyshabur et al., 2020), and even helped to design UDR (Duan et al., 2020), an unsupervised metric for model selection for VAEs.

The fact that good models are more similar to each other than bad ones in the context of classification (Morcos et al., 2018) generalised well to Unsupervised Disentanglement Ranking (UDR) (Rolinek et al., 2019; Duan et al., 2020). However, such a generalisation may not always be possible, and without sufficient evidence, it would be wise to expect substantial differences between the representations learned by supervised and unsupervised models. In this paper, our aim is to take a first step toward investigating the representational similarity of generative models by analysing the similarity scores obtained for a variety of VAEs learning disentangled representations, and by providing some insights into why various VAE-specific methods preventing posterior collapse (Bowman et al., 2016; He et al., 2019) or providing better reconstruction (Liu et al., 2021) are successful.

Our contributions are as follows:

(i) We provide the first experimental study of the representational similarity between VAEs, and have released more than 45 million similarity scores (`https://t.ly/0GLe3`[1]).

(ii) We have released the library created for this experiment (`https://t.ly/VMIm`). It can be reused with other similarity metrics or models for further research in the domain.

(iii) During our analysis, we found that (1) the encoder is learned before the decoder; (2) all the layers of the encoder, except the mean and variance layers, learn very similar representations regardless of the learning objective and regularisation strength used; and (3) linear CKA could be an efficient tool to track posterior collapse.

## 2 BACKGROUND

### 2.1 VARIATIONAL AUTOENCODERS

Variational Autoencoders (VAEs) (Kingma & Welling, 2014; Rezende & Mohamed, 2015) are deep probabilistic generative models based on variational inference. The encoder, $q_{\phi}(\mathbf{z}|\mathbf{x})$, maps some input $\mathbf{x}$ to a latent representation $\mathbf{z}$, which the decoder, $p_{\theta}(\mathbf{x}|\mathbf{z})$, uses to attempt to reconstruct $\mathbf{x}$. This can be optimised by maximising $\mathcal{L}$, the evidence lower bound (ELBO)

$$\mathcal{L}(\boldsymbol{\theta}, \boldsymbol{\phi}; \mathbf{x}) = \underbrace{\mathbb{E}_{q_{\phi}(\mathbf{z}|\mathbf{x})}[\log p_{\theta}(\mathbf{x}|\mathbf{z})]}_{\text{reconstruction term}} - \underbrace{D_{\mathrm{KL}}\left(q_{\phi}(\mathbf{z}|\mathbf{x})||p(\mathbf{z})\right)}_{\text{regularisation term}}, \tag{1}$$

where $p(\mathbf{z})$ is generally modelled as a multivariate Gaussian distribution $\mathcal{N}(0, \boldsymbol{I})$ to permit closed form computation of the regularisation term (Doersch, 2016). We refer to the regularisation term of Equation 1 as regularisation in the rest of the paper, and we do not tune any other forms of regularisation (e.g., L1, dropout). While our goal is not to study disentanglement, our experiments will focus on a range of VAEs designed to disentangle (Higgins et al., 2017; Chen et al., 2018; Burgess et al., 2018; Kumar et al., 2018) because they possess useful properties: (1) posterior collapse situations are easy to create; (2) these models are non-identifiable (Khemakhem et al., 2020), but we are interested in verifying if the representations learned still retain some linear relationship between models. We refer the reader to Appendix A for more details on these models.

**Polarised regime and posterior collapse** The polarised regime, also known as selective posterior collapse, is the ability of VAEs to "shut down" superfluous dimensions of their sampled latent representations while providing a high precision on the remaining ones (Rolinek et al., 2019; Dai et al., 2020). The existence of the polarised regime is a necessary condition for the VAEs to provide a good reconstruction (Dai & Wipf, 2018; Dai et al., 2020). However, when the weight on the regularisation term of the ELBO given in Equation 1 becomes too large, the representations collapse to the prior (Lucas et al., 2019a; Dai et al., 2020). Recently, Bonheme & Grzes (2021) have also shown that the passive variables, which are "shut down" during training, are very different in mean and sampled representations (see Appendix B). This indicates that representational similarity could be a valuable tool in the study of posterior collapse.

### 2.2 REPRESENTATIONAL SIMILARITY METRICS

Representational similarity metrics aim to compare the geometric similarity between two representations. In the context of deep learning, these representations correspond to $\mathbb{R}^{n \times p}$ matrices of activations, where $n$ is the number of data examples and $p$ the number of neurons in a layer. Such metrics can provide various information on deep neural networks (e.g., the training dynamics of neural networks, common and specialised layers between models).

**Centred Kernel Alignment** Centred Kernel Alignment (CKA) (Cortes et al., 2012; Cristianini et al., 2002) is a normalised version of the Hillbert-Schmit Independence Criterion (HSIC) (Gretton et al., 2005). As its name suggests, it measures the alignment between the $n \times n$ kernel matrices of two representations, and works well with linear kernels (Kornblith et al., 2019) for representational similarity of centred layer activations. We thus focus on the linear CKA, also known as

---

[1]Due to their size and to preserve anonymity, the 300 models trained for this paper will be released after the review.

RV-coefficient (Escoufier, 1973; Robert & Escoufier, 1976). Given the centered layer activations $\boldsymbol{X} \in \mathbb{R}^{n \times m}$ and $\boldsymbol{Y} \in \mathbb{R}^{n \times p}$ taken over $n$ data examples, linear CKA is defined as:

$$CKA(\boldsymbol{X}, \boldsymbol{Y}) = \frac{\|\boldsymbol{Y}^T \boldsymbol{X}\|_F^2}{\|\boldsymbol{X}^T \boldsymbol{X}\|_F \|\boldsymbol{Y}^T \boldsymbol{Y}\|_F}, \tag{2}$$

where $\|\cdot\|_F$ is the Frobenius norm. CKA is a generalisation of Pearson's correlation coefficient to higher dimensional representations (Escoufier, 1973; Robert & Escoufier, 1976) and can be seen as measuring the cosine between matrices (Josse & Holmes, 2016). It takes values between 0 (not similar) and 1 ($\boldsymbol{X} = \boldsymbol{Y}$). For conciseness, we will refer to linear CKA as CKA in the rest of this paper.

**Orthogonal Procrustes** The aim of orthogonal Procrustes (Schönemann, 1966) is to align a matrix $\boldsymbol{Y}$ to a matrix $\boldsymbol{X}$ using orthogonal transformations $\boldsymbol{Q}$ such that

$$\min_{\boldsymbol{Q}} \|\boldsymbol{X} - \boldsymbol{Y}\boldsymbol{Q}\|_F^2 \quad \text{s.t.} \quad \boldsymbol{Q}^T \boldsymbol{Q} = \boldsymbol{I}. \tag{3}$$

The Procrustes distance, $P_d$, is the difference remaining between $\boldsymbol{X}$ and $\boldsymbol{Y}$ when $\boldsymbol{Q}$ is optimal,

$$P_d(\boldsymbol{X}, \boldsymbol{Y}) = \|\boldsymbol{X}\|_F^2 + \|\boldsymbol{Y}\|_F^2 - 2\|\boldsymbol{Y}^T \boldsymbol{X}\|_*, \tag{4}$$

where $\|\cdot\|_*$ is the nuclear norm (see Golub & Van Loan (2013, pp. 327-328) for the full derivation from Equation 3 to Equation 4). To easily compare the results of Equation 4 with CKA, we first bound its results between 0 and 2 using normalised $\dot{\boldsymbol{X}}$ and $\dot{\boldsymbol{Y}}$, as detailed in Appendix C. Then, we transform the result to a similarity metric ranging from 0 (not similar) to 1 ($\boldsymbol{X} = \boldsymbol{Y}$),

$$P_s(\boldsymbol{X}, \boldsymbol{Y}) = 1 - \frac{1}{2}\left(\|\dot{\boldsymbol{X}}\|_F^2 + \|\dot{\boldsymbol{Y}}\|_F^2 - 2\|\dot{\boldsymbol{Y}}^T \dot{\boldsymbol{X}}\|_*\right). \tag{5}$$

We will refer to Equation 5 as Procrustes similarity in the following sections.

## 2.3 LIMITATIONS OF CKA AND PROCRUSTES SIMILARITIES

While CKA and Procrustes lead to accurate results in practice, they suffer from some limitations that need to be taken into account in our study. Before we discuss these limitations, we should clarify that, in the rest of this paper, $sim(\cdot, \cdot)$ represents a similarity metric in general, while $CKA(\cdot, \cdot)$ and $P_s(\cdot, \cdot)$ specifically refer to CKA and Procrustes similarities.

**Sensitivity to architectures** Maheswaranathan et al. (2019) have shown that similarity metrics comparing the geometry of representations were overly sensitive to differences in neural architectures. As CKA and Procrustes belong to this metrics family, we can expect them to underestimate the similarity between activations coming from layers of different type (e.g., convolutional and deconvolutional).

**Procrustes is sensitive to the number of data examples** As we may have representations with high dimensional features (e.g., activations of convolutional layers), we checked the impact of the number of data examples on CKA and Procrustes. To do so, we created four increasingly different matrices $\boldsymbol{A}, \boldsymbol{B}, \boldsymbol{C}$, and $\boldsymbol{D}$ with 50 features each: $\boldsymbol{B}$ retains 80% of $\boldsymbol{A}$'s features, $\boldsymbol{C}$ 50%, and $\boldsymbol{D}$ 0%. We then computed the similarity scores given by CKA and Procrustes while varying the number of data examples. As shown in Figures 1a and 1b, both metrics agree for $sim(\boldsymbol{A}, \boldsymbol{B})$ and $sim(\boldsymbol{A}, \boldsymbol{C})$, giving scores that are close to the fraction of common features between the two matrices. However, we can see in Figure 1c that Procrustes highly overestimates $sim(\boldsymbol{A}, \boldsymbol{D})$ while CKA scores rapidly drop.

**CKA ignores small changes in representations** When considering a sufficient number of data examples for both Procrustes and CKA, if two representations do not have dramatic differences (i.e., their 10% largest principal components are the same), CKA may overestimate similarity, while Procrustes remains stable, as observed by Ding et al. (2021).

**Similarity and disentanglement** It is important to keep in mind that CKA and Procrustes similarities are invariant to orthogonal transformation. Thus, they will consider a disentangled representation similar to a rotated (and possibly entangled) representation, as illustrated in Figure 2. Note that CKA and Procrustes are also invariant to isotropic scaling, though this does not affect disentanglement.

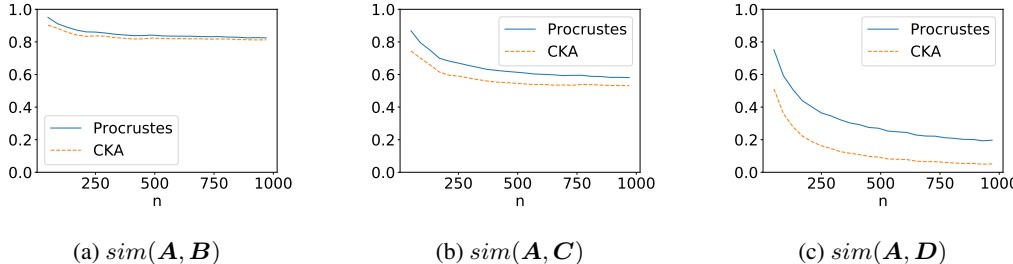

Figure 1: We compute CKA and Procrustes similarity scores with an increasing number of data examples $n$, and different similarity strength: $\boldsymbol{B}$ retains 80% of $\boldsymbol{A}$'s features, $\boldsymbol{C}$ 50%, and $\boldsymbol{D}$ 0%. Both metrics agree in (a) and (b), but Procrustes overestimates similarity in (c).

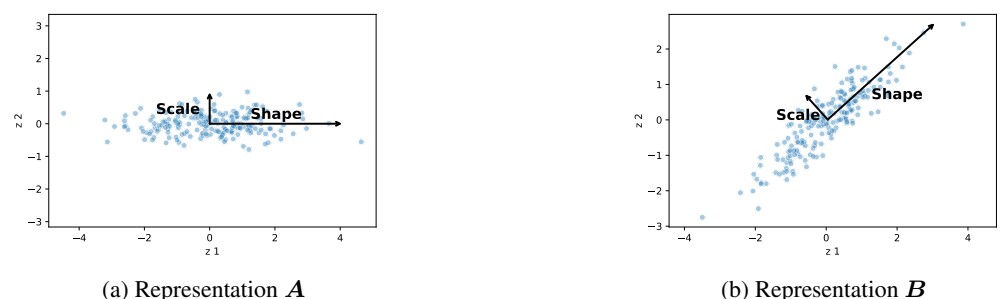

Figure 2: (a) shows a disentangled representation $\boldsymbol{A}$ where the latent dimensions $\boldsymbol{z}_1$ and $\boldsymbol{z}_2$ are aligned with some ground truth factors shape and scale. (b) shows a representation $\boldsymbol{B}$ which corresponds to an orthogonal transformation of $\boldsymbol{A}$. It is entangled as $\boldsymbol{z}_1$ and $\boldsymbol{z}_2$ are not aligned with the ground truth factors anymore, but $sim(\boldsymbol{A}, \boldsymbol{B}) = 1$ because Procrustes and CKA are invariant under orthogonal transformations.

**Ensuring accurate analysis**  Given the limitations previously mentioned, we take three remedial actions to guarantee that our analysis is as accurate as possible. Firstly, as both metrics will likely underestimate the similarity between different layer types, we will only discuss the variation of similarity when analysing such cases. For example, we will not compare $sim(\boldsymbol{A}, \boldsymbol{B})$ and $sim(\boldsymbol{A}, \boldsymbol{C})$ if $\boldsymbol{A}$ and $\boldsymbol{B}$ are convolutional layers but $\boldsymbol{C}$ is deconvolutional. We will nevertheless analyse the changes of $sim(\boldsymbol{A}, \boldsymbol{C})$ at different steps of training. Secondly, when both metrics disagree, we know that one of them is likely overestimating the similarity: Procrustes if the number of data examples is not sufficient, CKA if the difference between the two representations is not large enough. Thus, we will always use the smallest of the two results for our interpretations. Lastly, when we say that representations are similar, we mean that they are similar up to orthogonal rotations and isotropic scaling. Hence, two similar representations may not be equally disentangled.

## 3 EXPERIMENTAL SETUP

The goal of this experiment is to study the training dynamics of VAEs and the impact of initialisation, learning objectives, and regularisation on the representations learned by each layer. To do so, we measure the representational similarity of (i) One model at different epochs; (ii) Two models with the same learning objective and different regularisation strengths at the same epoch, and (iii) Two models with different learning objectives and equivalent regularisation strength at the same epoch.

**Learning objectives**  We will focus on learning objectives whose goal is to produce disentangled representations. Specifically, we use $\beta$-VAE (Higgins et al., 2017), $\beta$-TC VAE (Chen et al., 2018), Annealed VAE (Burgess et al., 2018), and DIP-VAE II (Kumar et al., 2018). A description of these methods can be found in Appendix A. To fairly provide complementary insights into previous observations of such models (Locatello et al., 2019b; Bonheme & Grzes, 2021), we will follow

the experimental design of Locatello et al. (2019b) regarding the architecture, learning objectives, and regularisation used. Moreover, `disentanglement lib`[2] will be used as a codebase for our experiment. The complete details are available in Appendix C.

**Datasets**   We use three datasets which, based on the results of Locatello et al. (2019b), are increasingly difficult for VAEs in terms of reconstruction loss: dSprites[3] (Higgins et al., 2017), cars3D (Reed et al., 2015), and smallNorb (LeCun et al., 2004).

**Training process**   We trained five models with different initialisations for 300,000 steps for each (learning objective, regularisation strength, dataset) triplet, and saved intermediate models to compare the similarity within individual models at different epochs. Appendix I explains our epoch selection methodology.

**Similarity measurement**   Given the computational complexity detailed below, for every dataset, we sampled 5,000 data examples, and we used them to compute all the similarity measurements. We compute the similarity scores between all pairs of layers of the different models following the different combinations outlined above. We will refer to the similarity scores of a group of one or more layers with itself as *self-similarity*. As Procrustes similarity takes significantly longer to compute compared to CKA (see below), we only used it to validate CKA results, restricting its usage to one dataset: cars3D. We obtained similar results for the two metrics on cars3D, thus we only reported CKA results in the main paper. Procrustes results can be found in Appendix D.

**Computational considerations**   Overall we trained 300 VAEs using 4 learning objectives, 5 different initialisations, 5 regularisation strengths, and 3 datasets, which took around 6,000 hours on an NVIDIA A100 GPU. We then computed the CKA scores for the 15 layer activations (plus the input) of each model combinations considered above at 5 different epochs, resulting in 470 million similarity scores and approximately 7,000 hours of computation on an Intel Xeon Gold 6136 CPU. As Procrustes is slowed down by the computation of the nuclear norm for high dimensional activations, the same number of similarity scores would have been prohibitively long to compute, requiring 30,000 hours on an NVIDIA A100 GPU. We thus only computed the Procrustes similarity for one dataset, reducing the computation time to 10,000 hours. Overall, based on the estimations of Lacoste et al. (2019), the computations done for this experiment amount to 2,200 Kg of $CO_2$, which corresponds to the $CO_2$ produced by one person over 5 months. To mitigate the negative environmental impact of our work, we released all our metric scores at `https://t.ly/0GLe3`. We hope that this will help to prevent unnecessary recomputation should others wish to reuse our results.

## 4   RESULTS

### 4.1   HOW ARE REPRESENTATIONS LEARNED AS TRAINING PROGRESSES?

In this section, we will analyse the learning dynamics of VAEs to answer objective (i) of Section 3. To monitor this, we will compare the representations learned by the first and last recorded snapshot of VAEs. Note that our choice of snapshots and snapshot frequency does not influence the results as verified in Appendices I and J. Note that when describing the results in heatmaps, we will refer to quadrants to indicate the similarity scores between all the representations of the encoder or decoder. This includes the off-diagonal CKA scores between layers of the same type.

**VAE learning is bottom-up**   As shown in Figure 3, the encoder is learned first, and the representations of its layers become similar to the input after a few epochs. The decoder then progressively learns representations that gradually become closer to the input. We observed a similar trend on fully-connected architectures, as reported in Appendix F. This result can explain why, when a decoder has access to the input (as it is the case in some autoregressive VAEs), it ignores the latent representations (Bowman et al., 2016; Li et al., 2019). Indeed, in this case the decoder is not constrained to wait for the encoder to converge before improving its reconstruction since it has direct

---

[2] `https://github.com/google-research/disentanglement_lib`
[3] Licensed under an Apache 2.0 licence.

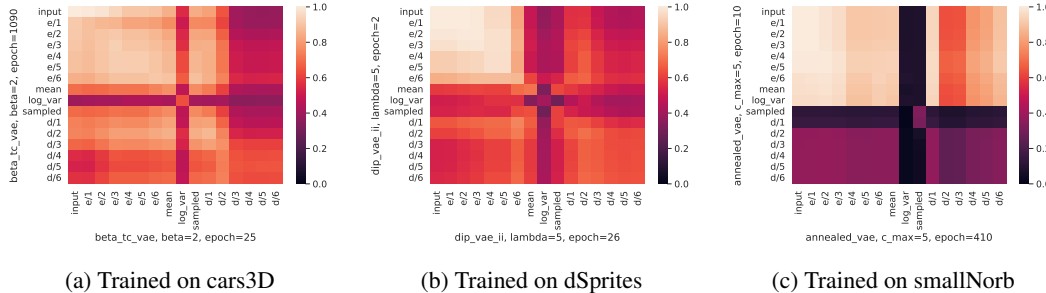

(a) Trained on cars3D     (b) Trained on dSprites     (c) Trained on smallNorb

Figure 3: (a) shows the CKA similarity scores of activations at epochs 25 and 1090 of $\beta$-TC VAE trained on cars3D with $\beta = 2$. (b) shows the CKA similarity scores of activations at epochs 2 and 26 of DIP-VAE II trained on dSprites with $\lambda = 5$. (c) shows the CKA similarity scores of activations at epochs 10 and 410 of Annealed VAE trained on smallNorb with $c_{max} = 5$. We can see that there is a high similarity between the representations learned by the encoder early in the training and after complete training (see the bright cells in the top-left quadrants in (a), (b), and (c)). However, the self-similarity between the decoder, mean and variance representations is lower, indicating some further changes in the representations after the first few epochs of training (see the dark cells in the bottom-right quadrant of parts (a), (b), and (c)). These results are averaged over 5 seeds.

access to the input sequences. In this paper, we used standard VAEs, where the decoder is not autoregressive and must rely on the latent representations as the sole source of information about the input. It thus needs to wait for the encoder to converge before being able to converge itself, effectively preventing posterior collapse when the regularisation is not too strong.

**Implications** The concurrent learning of encoder and decoder representations can be problematic. Indeed, if the encoder learns poor representations, the decoder will not be able to provide an accurate reconstruction. As a result, the encoder may struggle to provide better representations and this may lead to non-optimal results. We believe methods that facilitate the incremental learning of the encoder and decoder (e.g., by slowly increasing the complexity of the data, as in Progressive GANs (Karras et al., 2018)) as well as methods where the encoder is explicitly learned first (He et al., 2019) are promising ways to mitigate this issue. Moreover, the early convergence of the encoder can explain the success of "warm-up" methods such as annealing (Bowman et al., 2016) or aggressive inference (He et al., 2019) in the case where the encoder (quickly) converges to a bad local minima resulting in posterior collapse.

## 4.2 WHAT IS THE INFLUENCE OF THE HYPERPARAMETERS ON THE LEARNED REPRESENTATIONS?

So far, we have compared different snapshots of the same model. In this section, we will perform a fine-grained analysis of the impact of hyperparameters to answer objective (ii) of Section 3.

**Impact of regularisation** As discussed in (Bonheme & Grzes, 2021), passive variables have different values in the mean and sampled representations. This phenomenon is increasingly visible with higher regularisation, as more variables must become passive in order to lower the KL divergence (Rolinek et al., 2019; Dai et al., 2020). However, little is known about how this behaviour impacts the other layers of the encoder and decoder. As shown in Figure 4, the decoder representations change more than the encoder representations with the increased regularisation strength. This can be explained by Figure 5 where the sampled representations drift away from the input and mean representations, which is consistent with posterior collapse (Rolinek et al., 2019; Dai & Wipf, 2018; Dai et al., 2020). This is further confirmed by the fact that posterior collapse was already reported in (Bonheme & Grzes, 2021) for these configurations. Thus, our results indicate that CKA, which is quick to compute, could be a good tool to monitor the polarised regime and posterior collapse — its pathological counterpart — and differentiate well the two behaviours (see Appendix G). Interestingly, apart from the mean and variance, the representations learned by the encoder in Figure 4 stay

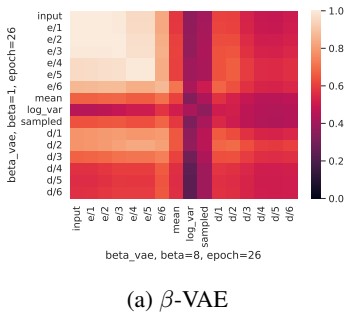

(a) $\beta$-VAE

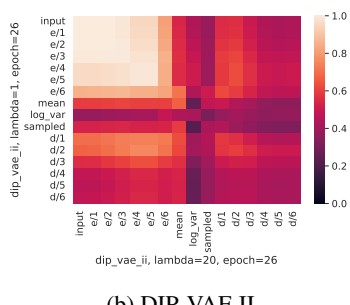

(b) DIP-VAE II

Figure 4: (a) shows the CKA similarity scores between the activations of two $\beta$-VAEs trained on dSprites with $\beta = 1$, and $\beta = 8$. (b) shows the CKA similarity scores between the activations of two DIP-VAE II trained on dSprites with $\lambda = 1$, and $\lambda = 20$. For (a) and (b), the activations are taken after complete training, and all the results are averaged over 5 seeds. In both figures, we can see that the self-similarity of decoder representations is low between models trained with low and high regularisation (dark cells in the bottom-right quadrants) while the encoder representations stay very similar (bright cells in the top-left quadrants), except for the mean, variance and sampled representations.

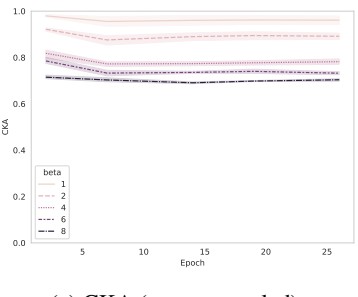

(a) CKA (mean, sampled)

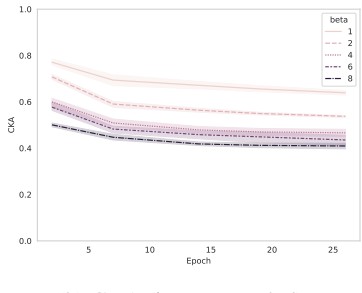

(b) CKA (input, sampled)

Figure 5: (a) and (b) show the CKA scores of $\beta$-VAEs trained on dSprites with $\beta$ from 1 to 8. (a) is the CKA between mean and sampled representations while (b) is between inputs and sampled representations. Both figures are consistent with posterior collapse caused by over-regularisation (Rolinek et al., 2019; Bonheme & Grzes, 2021) where the mean and sampled representations present a growing number of passive variables which, in the case of sampled representations, leads to high dissimilarity with the input in (b).

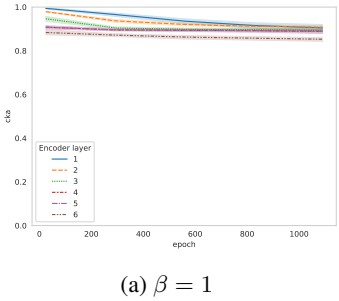

(a) $\beta = 1$

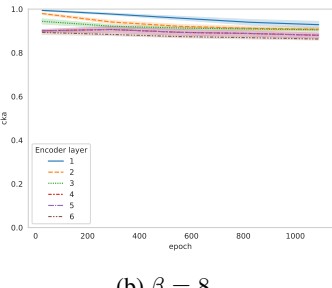

(b) $\beta = 8$

Figure 6: (a) shows the CKA scores between the inputs and the activations of the first 6 layers of the encoder of $\beta$-VAEs trained on cars3D with $\beta = 1$. (b) shows the scores between the same representations with $\beta = 8$. We can see that we have low variance across seeds (i.e., the shaded area corresponding to the variance interval around each line is small) and the encoder layers have an equivalent similarity with the input, regardless of the regularisation strength.

very similar. We can further see in Figure 6 that this is consistent across seeds and very different regularisation strengths, and can also be observed with fully-connected architectures in Appendix F.

**Implications**    The polarised regime (Rolinek et al., 2019; Zietlow et al., 2021; Dai et al., 2020) and posterior collapse (Dai & Wipf, 2018; Lucas et al., 2019a;b; Dai et al., 2020) do not seem to affect the representations learned by the encoder before the mean and variance layers (see Figure 6). Intuitively, this would imply that the encoder learns similar representations, regardless of the regularisation strength, and then "fine-tunes" them in its mean and variance layers.

### 4.3    What is the influence of the learning objectives on the learned representations?

We have seen in Section 4.2 that models with the same learning objective (e.g., $\beta$-VAE, DIP-VAE II or any other learning objective defined in Appendix A) but different regularisation were generally learning similar representations in the encoder except for the mean and variance layers. Starting from these layers and moving toward the decoder, the representational similarity decreases as the regularisation strength increases. Indeed, passive variables have different mean and variance representations than active variables (Rolinek et al., 2019; Bonheme & Grzes, 2021) and, as their number grows, so does the dissimilarity with the representations obtained at lower regularisation, which have fewer passive variables. Now we can wonder whether this pattern is also occuring in the context of objective (iii) of Section 3. For an equivalent regularisation strength but different learning objectives, do the models still learn similar representations in the layers of the encoder that are before the mean and variance layers?

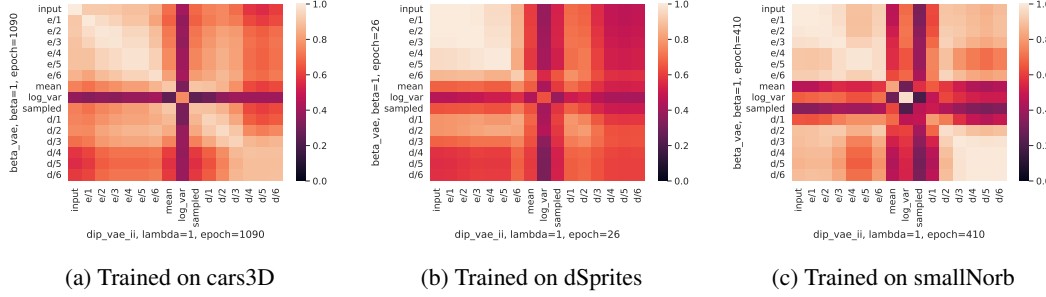

(a) Trained on cars3D          (b) Trained on dSprites          (c) Trained on smallNorb

Figure 7: (a) shows the CKA similarity scores of activations of $\beta$-VAE and DIP-VAE II trained on cars3D with $\beta = 1$, and $\lambda = 1$, respectively. (b) and (c) show the CKA similarity scores of the same learning objectives and regularisation strengths but trained on dSprites and smallNorb. These results are averaged over 5 seeds. We can see that the representational similarity of all the layers of the encoder (top-left quadrant) except mean and variance is very high (CKA stays close to 1). However, the mean, variance, sampled (center diagonal values), and decoder (bottom-right quadrants) representational similarity of different learning objectives seems to vary depending on the dataset. In (a) and (c) they have high similarity, while in (b) the similarity is lower.

**The representations learned by the early layers of the encoders are similar across learning objectives**    As seen in Section 4.2, the representations learned by the encoder are similar across hyperparameters, with mean and variance representations being less similar for high regularisation. In this experiment, we show that this observation also holds across learning objectives. We can see in Figure 7 that the representational similarity of different learning objectives with equivalent regularisation strength is quite high in all the encoder layers except mean and variance. This result is consistent across datasets, learning objectives, and well as architectures (see Appendix F), and the learned representations are also close to what is learned by a classifier with equivalent architecture (see Appendix H).

**The representations learned by the mean and variance layers can vary**    The similarity of the representations learned by the mean and variance layers (and consequently by the decoder) tends to vary across learning objectives and seems to be influenced by the dataset. Indeed, by looking at the center diagonal values of Figure 7, we can see that the similarity of the mean, variance, and

consequently sampled representations across different learning objectives is high for cars3D and smallNorb, but low for dSprites. This may indicate that, for a given dataset, different learning objectives can find different local optima, leading to lower similarity between mean and variance representations, and ultimately to different representations in the decoder. It seems that this phenomenon is very dataset-dependent, and is especially present in dSprites (see Figure 7b), which is one of the most used datasets for disentangled representation learning. The same phenomenon can be observed with fully-connected architectures in Appendix F.

**Implications** The representations learned by most of the encoder layers are very similar across learning objectives, indicating that the encoder may be learning some general features from the inputs, or the only features that can be learned when the decoder performs poorly. Indeed, since the decoder initially struggles to learn, the encoder may only be able to learn very general properties. The learning of general representations in early layers is consistent with Bansal et al. (2021), who, in the context of classification, observed that neural networks were learning similar representations regardless of the initialisation, architecture or learning objective used. As such, the encoder may be viewed as a feature extractor which is fine-tuned using a mean and variance layer to produce the sampled representations that will be used by the decoder. Our observations also suggest that some learning objectives may favour distinct local optima whose existence has been previously discussed (Alemi et al., 2017; Zietlow et al., 2021). Such a model-specific choice of local optima may explain why some learning objectives obtained better disentanglement scores than others on specific datasets but performed worse on others (Locatello et al., 2019b). Given that VAEs with non-conditional priors were shown to be non-identifiable (Khemakhem et al., 2020), it is very interesting to see that they still have similar encoders, and, for some datasets, similar decoders. Our results imply that for some datasets, different learning objectives can still provide decoder representations with strong linear relationships.

## 5 CONCLUSION

**Bottom-up learning and posterior collapse** As reported in Section 4.1, the encoder is learned before the decoder, which could indicate that the decoder struggles to converge before the mean and variance representations are learned. This would explain why one can observe posterior collapse in a setting where the decoder has access to the input, and thus can infer the mapping on its own (Bowman et al., 2016; Li et al., 2019).

**Different models encode similar representations** We have seen in Section 4 that the encoders, prior to their mean and variance layers, learned remarkably similar representations regardless of the initialisation, regularisation, and learning objective used to train the model. It is especially intriguing to see that even posterior collapse does not seem to affect these representations. The representational similarity of the mean, variance, and decoder representations across learning objectives generally vary depending on the dataset, indicating that different learning objectives may find different local optima for a given dataset. Note that this behaviour is more visible on some datasets (e.g., dSprites) than others (e.g., cars3D).

**Other applications** While our main focus was to compare similarity of models across a variety of settings, this study also demonstrated that CKA, whose computational cost is very low, can be an efficient tool to detect posterior collapse. Indeed, one can directly compare the similarity between mean and sampled representations, which strongly decreases as the number of collapsed variables grows (Bonheme & Grzes, 2021). We believe that this could be a complementary tool to the more costly mutual information generally used for such purpose.

**Limitations** We limited our study to similarity metrics that measure difference in the geometry of the representations. While this gave us compelling insights, these metrics have some limitations, discussed in Section 2.3, and may underestimate the similarity between layers with different architectures (Maheswaranathan et al., 2019). We believe that further research using dynamics-based metrics, such as fixed-point topology (Maheswaranathan et al., 2019), could provide additional insights into the representations learned by VAEs. It would also be beneficial to extend this study to other families of VAEs (van den Oord et al., 2017; Nalisnick & Smyth, 2017; Razavi et al., 2019; Vahdat & Kautz, 2020; Joo et al., 2020) to provide further insights into their various properties.

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

# A DISENTANGLED REPRESENTATION LEARNING

As mentioned in Section 2, we are interested in the family of methods modifying the weight on the regularisation term of Equation 1 to encourage disentanglement. In our paper, the term regularisation refers to the moderation of this parameter only. To achieve this, our experiment will focus on the models described below.

$\beta$-**VAE** The goal of this method (Higgins et al., 2017) is to penalise the regularisation term of Equation 1 by a factor $\beta > 1$, such that

$$\mathcal{L}(\boldsymbol{\theta}, \boldsymbol{\phi}; \mathbf{x}) = \mathbb{E}_{q_{\boldsymbol{\phi}}(\mathbf{z}|\mathbf{x})} \left[\log p_{\boldsymbol{\theta}}(\mathbf{x}|\mathbf{z})\right] - \beta D_{\mathrm{KL}}(q_{\boldsymbol{\phi}}(\mathbf{z}|\mathbf{x})||p(\mathbf{z})). \tag{6}$$

**Annealed VAE** Burgess et al. (2018) proposed to gradually increase the encoding capacity of the network during the training process. The goal is to progressively learn latent variables by decreasing order of importance. This leads to the following objective, where C is a parameter that can be understood as a channel capacity and $\gamma$ is a hyper-parameter penalising the divergence, similarly to $\beta$ in $\beta$-VAE:

$$\mathcal{L}(\boldsymbol{\theta}, \boldsymbol{\phi}; \mathbf{x}) = \mathbb{E}_{q_{\boldsymbol{\phi}}(\mathbf{z}|\mathbf{x})} \left[\log p_{\boldsymbol{\theta}}(\mathbf{x}|\mathbf{z})\right] - \gamma \left|D_{\mathrm{KL}}(q_{\boldsymbol{\phi}}(\mathbf{z}|\mathbf{x})||p(\mathbf{z})) - \mathrm{C}\right|. \tag{7}$$

As the training progresses, the channel capacity C is increased, going from zero to its maximum channel capacity $\mathrm{C}_{\max}$ and allowing a higher value of the KL divergence term. VAEs that use Equation 7 as a learning objective are referred to as Annealed VAEs in this paper.

$\beta$-**TC VAE** Chen et al. (2018) argued that only the distance between the estimated latent factors and the prior should be penalised to encourage disentanglement, such that

$$\mathcal{L}(\boldsymbol{\theta}, \boldsymbol{\phi}; \mathbf{x}) = \mathbb{E}_{p(\mathbf{x})} \left[\mathbb{E}_{q_{\boldsymbol{\phi}}(\mathbf{z}|\mathbf{x})} \left[\log p_{\boldsymbol{\theta}}(\mathbf{x}|\mathbf{z})\right] - D_{\mathrm{KL}}\left(q_{\boldsymbol{\phi}}(\mathbf{z}|\mathbf{x})||p(\mathbf{z})\right)\right] - \lambda D_{\mathrm{KL}}(q_{\boldsymbol{\phi}}(\mathbf{z})||p(\mathbf{z})). \tag{8}$$

Here, $D_{\mathrm{KL}}(q_{\boldsymbol{\phi}}(\mathbf{z})||p(\mathbf{z}))$ is approximated by penalising the dependencies between the dimensions of $q_{\boldsymbol{\phi}}(\mathbf{z})$:

$$\mathcal{L}(\boldsymbol{\theta}, \boldsymbol{\phi}; \mathbf{x}) \approx \frac{1}{n} \sum_{i=1}^{n} \left[\mathbb{E}_{q_{\boldsymbol{\phi}}(\mathbf{z}|\mathbf{x})} \left[\log p_{\boldsymbol{\theta}}(\boldsymbol{x}_i|\mathbf{z})\right] - D_{\mathrm{KL}}\left(q_{\boldsymbol{\phi}}(\mathbf{z}|\boldsymbol{x}_i)||p(\mathbf{z})\right)\right] - \lambda \underbrace{D_{\mathrm{KL}}(q_{\boldsymbol{\phi}}(\mathbf{z})|| \prod_{j=1}^{\mathrm{D}} q_{\boldsymbol{\phi}}(\mathbf{z}_j))}_{\text{total correlation}}. \tag{9}$$

The total correlation of Equation 9 is then approximated over a mini-batch of samples $\{\boldsymbol{x}_1, \ldots, \boldsymbol{x}_M\}$ as follows:

$$\mathbb{E}_{q_{\boldsymbol{\phi}}(\mathbf{z})}[\log q_{\boldsymbol{\phi}}(\mathbf{z})] \approx \frac{1}{M} \sum_{i=1}^{M} \left(\log \frac{1}{NM} \sum_{j=1}^{M} q_{\boldsymbol{\phi}}(z(\boldsymbol{x}_i)|\boldsymbol{x}_j)\right),$$

where $z(\boldsymbol{x}_i)$ is a sample from $q_{\boldsymbol{\phi}}(\mathbf{z}|\boldsymbol{x}_i)$, $M$ is the number of samples in the mini-batch, and $N$ the total number of input examples. $\mathbb{E}_{q_{\boldsymbol{\phi}}(\mathbf{z}_i)}[\log q_{\boldsymbol{\phi}}(\mathbf{z}_i)]$ can be computed in a similar way.

**DIP-VAE** Similarly to Chen et al. (2018), Kumar et al. (2018) proposed to regularise the distance between $q_{\boldsymbol{\phi}}(\mathbf{z})$ and $p(\mathbf{z})$ using Equation 8. The main difference is that here $D_{\mathrm{KL}}(q_{\boldsymbol{\phi}}(\mathbf{z})||p(\mathbf{z}))$ is measured by matching the moments of the learned distribution $q_{\boldsymbol{\phi}}(\mathbf{z})$ and its prior $p(\mathbf{z})$. The second moment of the learned distribution is given by

$$\mathrm{Cov}_{q_{\boldsymbol{\phi}}(\mathbf{z})}[\mathbf{z}] = \mathrm{Cov}_{p(\mathbf{x})} \left[\mu_{\boldsymbol{\phi}}(\mathbf{x})\right] + \mathbb{E}_{p(\mathbf{x})} \left[\Sigma_{\boldsymbol{\phi}}(\mathbf{x})\right]. \tag{10}$$

DIP-VAE II penalises both terms of Equation 10 such that

$$\lambda D_{\mathrm{KL}}(q_{\boldsymbol{\phi}}(\mathbf{z})||p(\mathbf{z})) = \lambda_{od} \sum_{i \neq j} \left(\mathrm{Cov}_{q_{\boldsymbol{\phi}}(\mathbf{z})}[\mathbf{z}]\right)_{ij}^2 + \lambda_d \sum_i \left(\mathrm{Cov}_{q_{\boldsymbol{\phi}}(\mathbf{z})}[\mathbf{z}]_{ii} - 1\right)^2,$$

where $\lambda_d$ and $\lambda_{od}$ are the penalisation terms for the diagonal and off-diagonal values respectively.

# B  ADDITIONAL DETAILS ON MEAN, VARIANCE AND SAMPLED REPRESENTATIONS

This section presents a concise illustration of what mean, variance and sampled representations are. As shown in Figure 8, the mean, variance and sampled representations are the last 3 layers of the encoder, where the sampled representation, $z$, is the input of the decoder. These representations, specific to VAEs, influence the models' behaviour that can be quite different from other deep learning models, as shown by research on polarised regime and posterior collapse, for example.

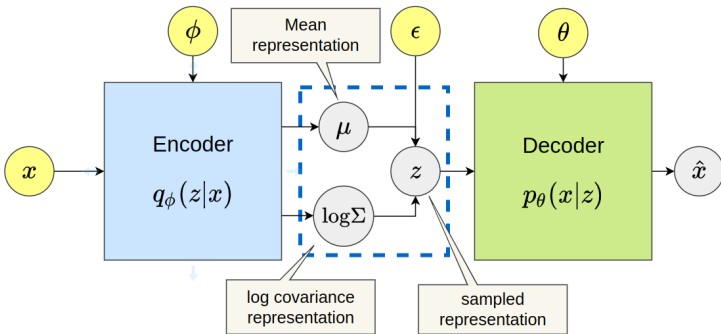

Figure 8: The structure of a VAE

# C  EXPERIMENTAL SETUP

To facilitate the reproducibility of our experiment, we detail below the Procrustes normalisation process and the configuration used for model training.

**Procrustes normalisation**    Similarly to Ding et al. (2021), given an activation matrix $\boldsymbol{X} \in \mathbb{R}^{n \times m}$ containing $n$ samples and $m$ features, we compute the vector $\bar{\boldsymbol{x}} \in \mathbb{R}^m$ containing the mean values of the columns of $\boldsymbol{X}$. Using the outer product $\otimes$, we get $\bar{\boldsymbol{X}} = \mathbf{1}_n \otimes \bar{\boldsymbol{x}}$, where $\mathbf{1}_n \in \mathbb{R}^n$ is a vector of ones and $\bar{\boldsymbol{X}} \in \mathbb{R}^{n \times m}$. We then normalise $\boldsymbol{X}$ such that

$$\dot{\boldsymbol{X}} = \frac{\boldsymbol{X} - \bar{\boldsymbol{X}}}{\|\boldsymbol{X} - \bar{\boldsymbol{X}}\|_F}. \tag{11}$$

As the Frobenius norm of $\dot{\boldsymbol{X}}$ and $\dot{\boldsymbol{Y}}$ is 1, and $\|\dot{\boldsymbol{Y}}^T \dot{\boldsymbol{X}}\|_*$ is always positive (1 when $\dot{\boldsymbol{X}} = \dot{\boldsymbol{Y}}$, smaller otherwise), Equation 4 lies in $[0, 2]$, and Equation 5 in $[0, 1]$.

**VAE training**    Our implementation uses the same hyperparameters as Locatello et al. (2019b), and the details are listed in Table 1 and 2. We reimplemented Locatello et al. (2019b) code base, designed for Tensorflow 1, in Tensorflow 2 using Keras. The model architecture used is also identical, as described in Table 3. Each model is trained 5 times, on seeded runs with seed values from 0 to 4. Intermediate models are saved every 1,000 steps for smallNorb, 6,000 steps for cars3D and 11,520 steps for dSprites. Every image input is normalised to have pixel values between 0 and 1.

For the fully-connected models presented in Appendix F, we used the same architecture and hyperparameters as those implemented in `disentanglement_lib` of Locatello et al. (2019b), and the details are presented in Table 4 and 5.

# D  CONSISTENCY OF THE RESULTS WITH PROCRUSTES SIMILARITY

As mentioned in Section 3, in this section we provide a comparison between the CKA scores reported in the main paper, and the Procrustes scores for the cars3D dataset. We can see in Figures 9 to 11 that Procrustes and CKA provide similar results. Figures 9 and 10 show that Procrustes tends to overestimate the similarity between high-dimensional inputs, as mentioned in Section 2.3 (recall

Table 1: Shared hyperparameters

| Parameter | Value |
|---|---|
| Batch size | 64 |
| Latent space dimension | 10 |
| Optimizer | Adam |
| Adam: $\beta_1$ | 0.9 |
| Adam: $\beta_2$ | 0.999 |
| Adam: $\epsilon$ | 1e-8 |
| Adam: learning rate | 0.0001 |
| Reconstruction loss | Bernoulli |
| Training steps | 300,000 |
| Intermediate model saving | every 6K steps |
| Train/test split | 90/10 |

Table 2: Model-specific hyperparameters

| Model | Parameter | Value |
|---|---|---|
| $\beta$-VAE | $\beta$ | [1, 2, 4, 6, 8] |
| $\beta$-TC VAE | $\beta$ | [1, 2, 4, 6, 8] |
| DIP-VAE II | $\lambda_{od}$ | [1, 2, 5, 10, 20] |
| | $\lambda_d$ | $\lambda_{od}$ |
| Annealed VAE | $C_{max}$ | [5, 10, 25, 50, 75] |
| | $\gamma$ | 1,000 |
| | iteration threshold | 100,000 |

Table 3: Shared architecture

| Encoder | Decoder |
|---|---|
| Input: $\mathbb{R}^{64 \times 63 \times channels}$ | $\mathbb{R}^{10}$ |
| Conv, kernel=4×4, filters=32, activation=ReLU, strides=2 | FC, output shape=256, activation=ReLU |
| Conv, kernel=4×4, filters=32, activation=ReLU, strides=2 | FC, output shape=4x4x64, activation=ReLU |
| Conv, kernel=4×4, filters=64, activation=ReLU, strides=2 | Deconv, kernel=4×4, filters=64, activation=ReLU, strides=2 |
| Conv, kernel=4×4, filters=64, activation=ReLU, strides=2 | Deconv, kernel=4×4, filters=32, activation=ReLU, strides=2 |
| FC, output shape=256, activation=ReLU, strides=2 | Deconv, kernel=4×4, filters=32, activation=ReLU, strides=2 |
| FC, output shape=2x10 | Deconv, kernel=4×4, filters=channels, activation=ReLU, strides=2 |

Table 4: Fully-connected architecture

| Encoder | Decoder |
|---|---|
| Input: $\mathbb{R}^{64 \times 63 \times channels}$ | $\mathbb{R}^{10}$ |
| FC, output shape=1200, activation=ReLU | FC, output shape=256, activation=tanh |
| FC, output shape=1200, activation=ReLU | FC, output shape=1200, activation=tanh |
| FC, output shape=2x10 | FC, output shape=1200, activation=tanh |

the example given in Figure 1). In Figure 11, we observe a slightly lower similarity with Procrustes than CKA on the $5^{th}$ and $6^{th}$ layers of the encoder, indicating that some small changes in the representations may have been underestimated by CKA, as discussed in Section 2.3 and by Ding et al. (2021). Note that the difference between the CKA and Procrustes similarity scores in Figure 11 remains very small (around 0.1) indicating consistent results between both metrics.

Table 5: Hyperparameters of fully-connected models

| Model | Parameter | Value |
|---|---|---|
| $\beta$-VAE | $\beta$ | [1, 8, 16] |
| $\beta$-TC VAE | $\beta$ | [2] |
| DIP-VAE II | $\lambda_{od}$ | [1, 20, 50] |
| | $\lambda_d$ | $\lambda_{od}$ |
| Annealed VAE | $C_{max}$ | [5] |
| | $\gamma$ | 1,000 |
| | iteration threshold | 100,000 |

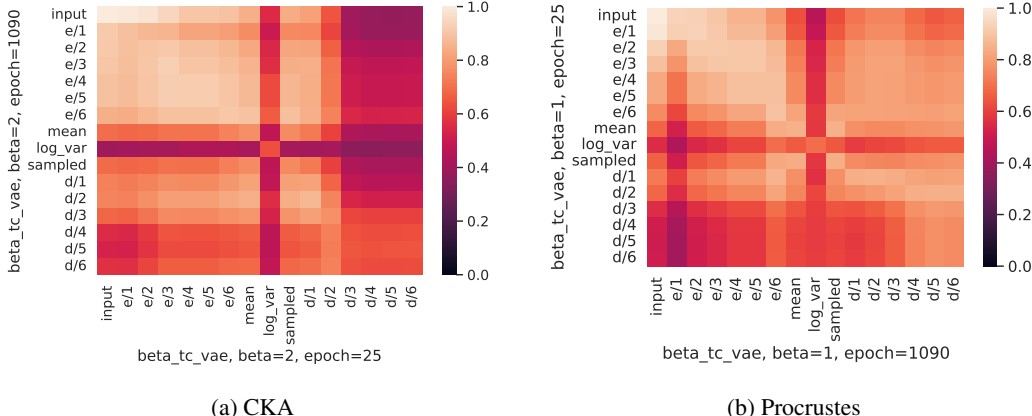

(a) CKA          (b) Procrustes

Figure 9: As reported in Figure 3a of the main paper, (a) shows the CKA similarity scores of activations at epochs 25 and 1090 of $\beta$-TC VAE trained on cars3D with $\beta = 2$. (b) shows the Procrustes similarity scores of the same configuration. We observe the same trend with both metrics with Procrustes slightly overestimating the similarity between high dimensional activations (bottom-right quadrants), which agrees with the properties of the Procrustes similarity reported in Section 2.3.

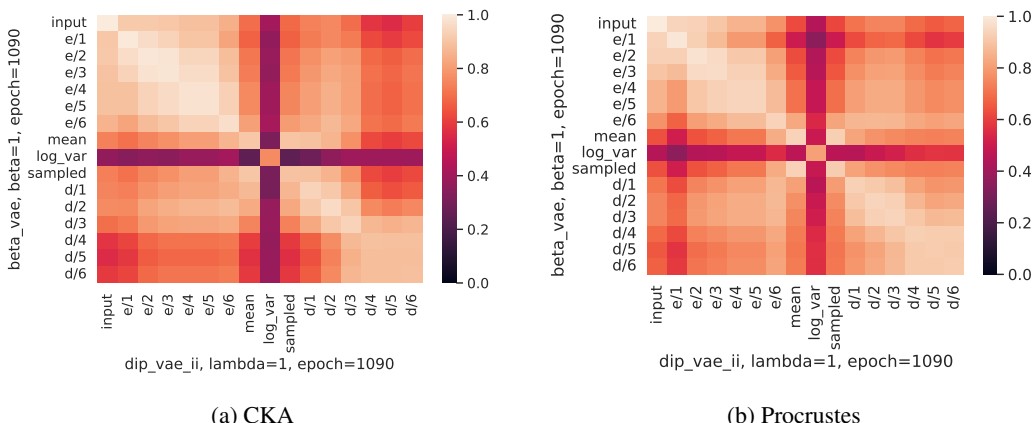

(a) CKA          (b) Procrustes

Figure 10: As in Figure 7a, (a) shows the CKA similarity scores of activations of $\beta$-VAE and DIP-VAE II trained on cars3D with $\beta = 1$, and $\lambda = 1$, respectively. (b) shows the Procrustes similarity scores using the same configuration. We observe the same trend with both metrics with Procrustes slightly overestimating the similarity between high dimensional activations (bottom-right quadrants) (c.f. Section 2.3).

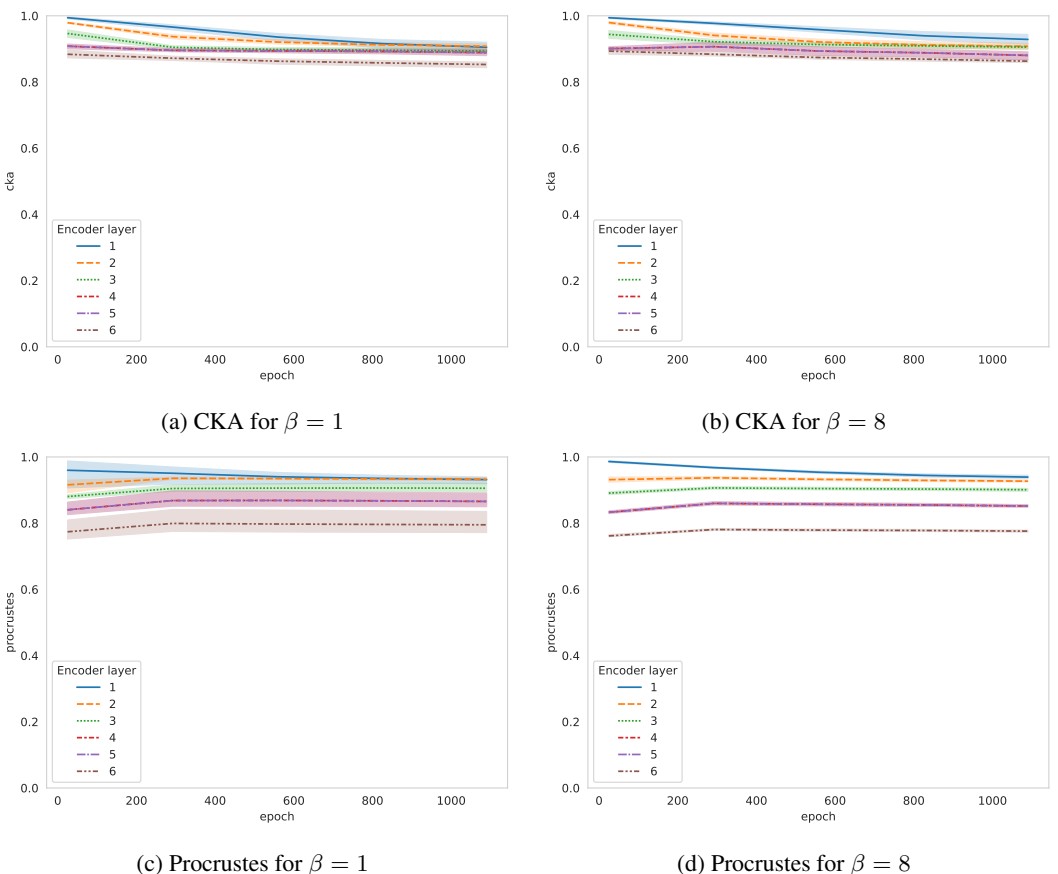

(a) CKA for $\beta = 1$

(b) CKA for $\beta = 8$

(c) Procrustes for $\beta = 1$

(d) Procrustes for $\beta = 8$

Figure 11: (a) shows the CKA scores between the inputs and the activations of the first 6 layers of the encoder of a $\beta$-VAE trained on cars3D with $\beta = 1$. (b) shows the scores between the same representations with $\beta = 8$. (a) and (b) are discussed in Figure 6 of the main paper. (c) and (d) are the Procrustes scores of the same configurations. We observe the same trend for both metrics with more variance in (c) for Procrustes with $\beta = 1$. Procrustes also displays a slightly lower similarity for layers 4 to 6 of the encoder, possibly due to changes in the representation that are underestimated by CKA (c.f. Section 2.3).

## E  RESOURCES

As mentioned in Sections 1 and 3, we released the code of our experiment, the pre-trained models and similarity scores:

- The similarity scores can be downloaded from an anonymous Google account using the following tiny URL `https://t.ly/0GLe3`

- The code can also be downloaded from an anonymous Google account using another tiny URL `https://t.ly/VMIm`

- Our pre-trained models are large (around 80 GB in total), and it was not feasible to make them available to the reviewers using an anonymous link. The URL to the models will, however, be available in the non-anonymised version of this paper.

Note that the 300 VAE models released correspond to models trained with:

- 4 different learning objectives,
- 5 initialisations,
- 3 datasets,
- 5 regularisation strengths.

## F  CKA ON FULLY-CONNECTED ARCHITECTURES

In order to assess the generalisability of our findings, we have repeated our observations on the fully-connected VAEs that are described in Appendix C. We can see in Figures 12, 13, and 14 that the same general trend as for the convolutional architectures can be identified (see Figures 3c, 4a, and 7 of  Sections 4.1, 4.2 and 4.3 for a comparison with convolutional networks).

**Learning in fully-connected VAEs is also bottom-up**   We can see in Figure 12 that, similarly to the convolutional architectures shown in Figure 3c, the encoder is learned early in the training process. Indeed between epochs 1 and 10, the encoder representations become highly similar to the representations of the fully trained model (see Figures 12a and 12b). The decoder is then learned with its representational similarity with the fully trained decoder raising after epoch 10 (see Figure 12c).

**Impact of regularisation**   As in convolutional architectures shown in Figure 4a, the variance and sampled representations retain little similarity with the encoder representations in the case of posterior collapse, as shown in Figure 13. Interestingly, in fully-connected architectures the decoder retains more similarity with its less regularised version than in convolutional architectures, despite suffering from poor reconstruction when heavily regularised. Thus, CKA of the representations of fully-connected decoders may not be a good predictor of reconstruction quality. Despite this difference, it can still be used to monitor posterior collapse with fully-connected architecture by relying on the similarity scores between the encoder representations, and the mean, variance and sampled representations. This property is consistent between both architectures.

**Impact of learning objective**   Figure 14 provides results similar to the convolutional VAEs observed in Figure 7, with a very high similarity between encoder layers learned from different learning objectives (see diagonal values of the upper-left quadrant). Here again, the representational similarity of the decoder seems to vary depending on the dataset, even though this is less marked than for convolutional architectures. We can also see that the representational similarity between different layers of the encoder vary depending on the dataset, which was less visible in convolutional architectures. For example, the similarity between the first and subsequent layers of the encoder in smallNorb is much lower in fully-connected VAEs. Given that smallNorb is a hard dataset to learn for VAEs (Locatello et al., 2019b), one could hypothesise that the encoder of fully-connected VAE, being less powerful, is unable to retain as much information as its convolutional counterpart, leading to lower similarity scores with the representations of the first encoder layer.

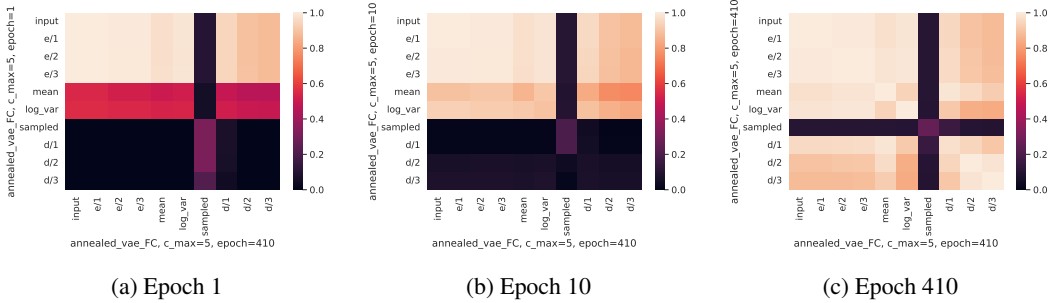

(a) Epoch 1            (b) Epoch 10            (c) Epoch 410

Figure 12: (a), (b), and (c) show CKA scores between a fully-trained fully-connected Annealed VAE and a fully-connected Annealed VAE trained for 1, 10 and 410 epochs, respectively. All the models are trained on smallNorb and the results are averaged over 5 seeds. Similarity to Figure 3c of Section 4.1, we can see that there is a high similarity between the representations learned by the encoder early in the training and after complete training (see the bright cells in the top-left quadrants in (a), (b), and (c)). The mean and variance representations similarity with a fully trained model increase after a few more epochs (the purple line in the middle disappear between (a) and (b)), and finally the decoder is learned (see bright cells in bottom-right quadrant of (c)).

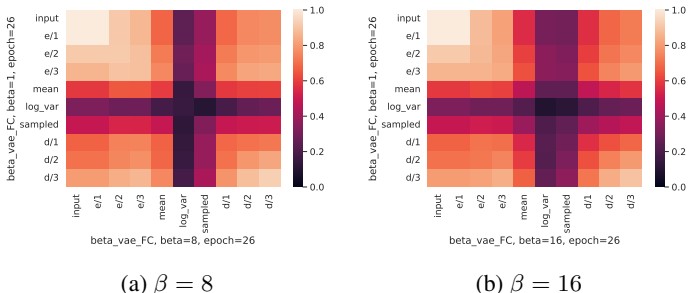

(a) $\beta = 8$            (b) $\beta = 16$

Figure 13: (a) and (b) show the representational similarity between fully-connected $\beta$-VAEs trained with $\beta = 1$, and fully-connected $\beta$-VAEs trained with $\beta = 8$ and $\beta = 16$, respectively. All models are trained on dSprites and the scores are averaged over 5 seeds. In both figures, the encoder representations stay very similar (bright cells in the top-left quadrants), except for the mean, variance and sampled representations. While the variance representation is increasingly different as we increase $\beta$, the decoder does not show the dramatic dissimilarity observed in convolutional architectures in Figure 4a.

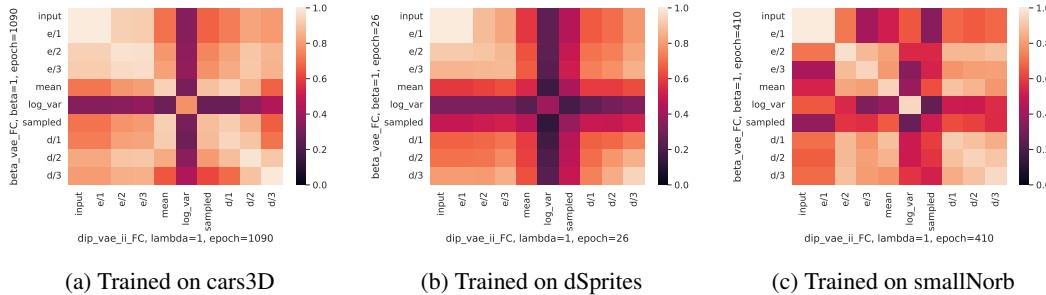

(a) Trained on cars3D  (b) Trained on dSprites  (c) Trained on smallNorb

Figure 14: (a) shows the CKA similarity scores of activations of $\beta$-VAE and DIP-VAE II with fully-connected architectures trained on cars3D with $\beta = 1$, and $\lambda = 1$, respectively. (b) and (c) show the CKA similarity scores of the same learning objectives and regularisation strengths but trained on dSprites and smallNorb. All the results are averaged over 5 seeds. We can see that the representational similarity of all the layers of the encoder (top-left quadrant) except mean and variance is high (CKA $\geqslant 0.8$). However, as in Figure 7, the mean, variance, sampled (center diagonal values), and decoder (bottom-right quadrants) representational similarity of different learning objectives seems to vary depending on the dataset. In (a) and (c) they have high similarity, while in (b) the similarity is lower. Moreover, in (c) the input and first layer of the encoder are quite distinct from the other representations, which was not the case in convolutional architectures.

## G  How well does CKA distinguish polarised regime from posterior collapse?

In Section 4.2, we stated that CKA could be a useful tool to monitor posterior collapse. Indeed, we have seen in Section 2.1 that the polarised regime can be viewed as a "healthy" version of the posterior collapse where the only latent variables that are not needed become passive, while posterior collapse will result in making most of the latent variables passive, preventing the decoder from providing a good reconstruction. In both cases, passive variables are close to $\mathcal{N}(0, 1)$ in the sampled representation. For this to be possible, the mean representation of passive variables is always very close to 0, and the variance representation to 1, so that the sampling process results in $\boldsymbol{z}_i \approx \epsilon$ instead of $\boldsymbol{z}_i = \mu_i + \sigma_i \epsilon$ where $\epsilon \sim \mathcal{N}(0, 1)$. While passive variables exhibit differences between the mean and sampled representations, the active variables are very similar in mean and sampled representations. Indeed, in this case, the variance is very low, leading to $z_i \approx \mu_i$.

Keeping in mind that linear CKA is a generalisation of the correlation coefficient to matrices, if there is any linear relationship between two representations, CKA will be high and the stronger this relationship is, the higher CKA will be.

Because mean and sampled representations have similar active variables but different passive variables, the presence of active variables will result in higher CKA scores. Thus, mean and sampled representation will have higher CKA in polarised regime (where they contain many active variables) than during posterior collapse (where there are none or very few active variables). However, one can wonder whether CKA can lead to false positives when VAEs contain passive variables for non-pathological reasons. For example, due to the polarised regime, if one provides more latent variables than needed by the VAEs, some of the variables will be collapsed to reduce the KL divergence. As in posterior collapse, the decoder will ignore these passive variables. However, contrary to collapsed models, when passive variables are a result of the polarised regime, the decoder will still have access to meaningful information and will be able to correctly reconstruct the image, learning similar representations as a good model with fewer latent variables. We can see in Figure 15 that, in opposition to posterior collapse, the variance and sampled representations retain much higher similarity scores with the representations learned by other layers. Thus, one can differentiate between the two scenarios using CKA. Given that the CKA scores for the variance and sampled representations vary similarly in fully-connected architectures, CKA seems to consistently be a good predictor of posterior collapse across learning objectives and architectures while being robust to false positives. While one could be tempted to monitor posterior collapse using the changes of similarity scores in the decoder, we have seen in Appendix F that the fully-connected decoders could retain a relatively high

similarity in the case of posterior collapse. Thus, we recommend relying on the CKA scores of the mean, variance and sampled representations for a better robustness across architectures.

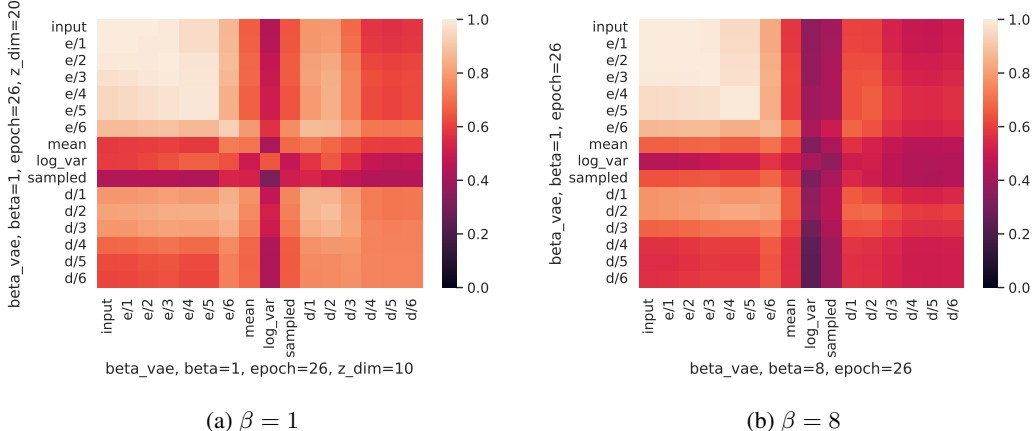

(a) $\beta = 1$                 (b) $\beta = 8$

Figure 15: (a) shows the CKA similarity scores between the activations of two $\beta$-VAEs trained on dSprites with $\beta = 1$, the first with 10 latent variables, and the second with 20. (b) shows the CKA similarity scores between the activations of two $\beta$-VAEs with 10 latent variables trained on dSprites with $\beta = 1$, and $\beta = 8$. For (a) and (b), the activations are taken after complete training, and all the results are averaged over 5 seeds. We can see that we retain a higher similarity in the decoder representations in the case of polarised regime (bottom-right quadrant in (a)) than posterior collapse (see the darker bottom-right quadrant in (b)). The representational similarity of the mean, variance and sampled representations is also higher in the case of polarised regime than posterior collapse.

## H    HOW SIMILAR ARE THE REPRESENTATIONS LEARNED BY ENCODERS AND CLASSIFIERS?

To compare VAEs with classifiers, we used the convolutional architecture of an encoder for classification, replacing the mean and variance layers by the final classifier layers. As shown in Figure 16, we obtain a high representational similarity when comparing VAEs and classifiers indicating, consistently with the observations of Yosinski et al. (2015), that classifiers seem to learn generative features. This explains why encoders based on pre-trained classifier architectures such as VGG have empirically demonstrated good performances (Liu et al., 2021) and also suggests that the weights of the pre-trained architecture could be used as-is without further updates. While using pre-trained encoders may be beneficial in the context of transfer learning, domain adaptation (Pan & Yang, 2009), or simply reconstruction quality (Liu et al., 2021), one should not expect a dramatic improvement of the training time given that the encoder is learned very early during the training (see Section 4.1).

## I    REPRESENTATIONAL SIMILARITY OF VAES AT DIFFERENT EPOCHS

The results obtained in Section 4.1 have shown a high similarity between the encoders at an early stage of training and fully trained. One can wonder whether these results are influenced by the choice of epochs used in Figure 3. After explaining our epoch selection process, we show below that it does not influence our results, which are consistent over snapshots taken at different stages of training.

**Epoch selection**    For dSprites, we took snapshots of the models at each epoch, but for cars3D and smallNorb, which both train for a higher number of epochs, it was not feasible computationally to calculate the CKA between every epoch. We thus saved models trained on smallNorb every 10 epochs, and models trained on cars3D every 25 epochs. Consequently, the epochs chosen to represent the early training stage in Section 4 is always the first snapshot taken for each model. Below, we preform additional experiments with a broader range of epoch numbers to show that the results are consistent with our findings in the main paper, and they do not depend on specific epochs.

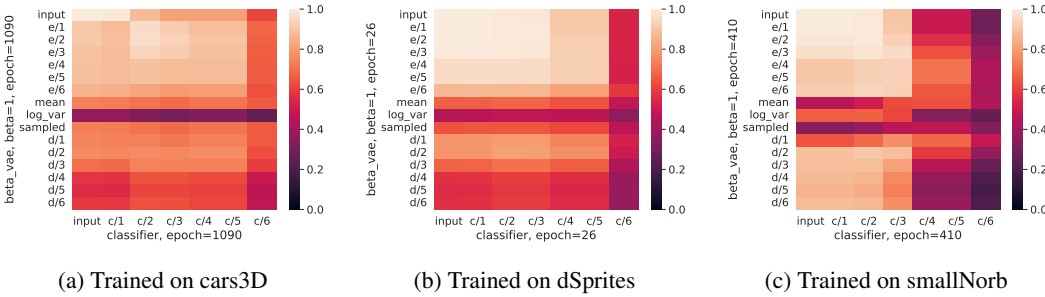

(a) Trained on cars3D      (b) Trained on dSprites      (c) Trained on smallNorb

Figure 16: (a) shows the CKA similarity scores of activations of a classifier and a $\beta$-VAE trained on cars3D with $\beta = 1$. (b) and (c) show the CKA similarity scores of the same learning objectives and regularisation strengths but trained on dSprites and smallNorb. All the results are averaged over 5 seeds. We can see that the representational similarity between the layers of the classifiers and of the encoder (top-left quadrant) except mean and variance is very high (CKA stays close to 1). However, the mean, variance, sampled representations (center diagonal values) are different from the representations learned by the classifier.

**Similarity changes over multiple epochs**  In Figures 17, 18, and 19, we can observe the same trend of learning phases as in Figure 3. First, the encoder is learned, as shown by the high representational similarity of the upper-left quadrant of Figures 17a, 18a, and 19a. Then, the decoder is learned, as shown by the increased representational similarity of the bottom-right quadrant of Figures 17b, 18b, and 19b. Finally, further small refinements of the encoder and decoder representations take place in the remaining training time, as shown by the slight increase of representational similarity in Figures 17c, 18c, and 19c, and Figures 17d, 18d, and 19d.

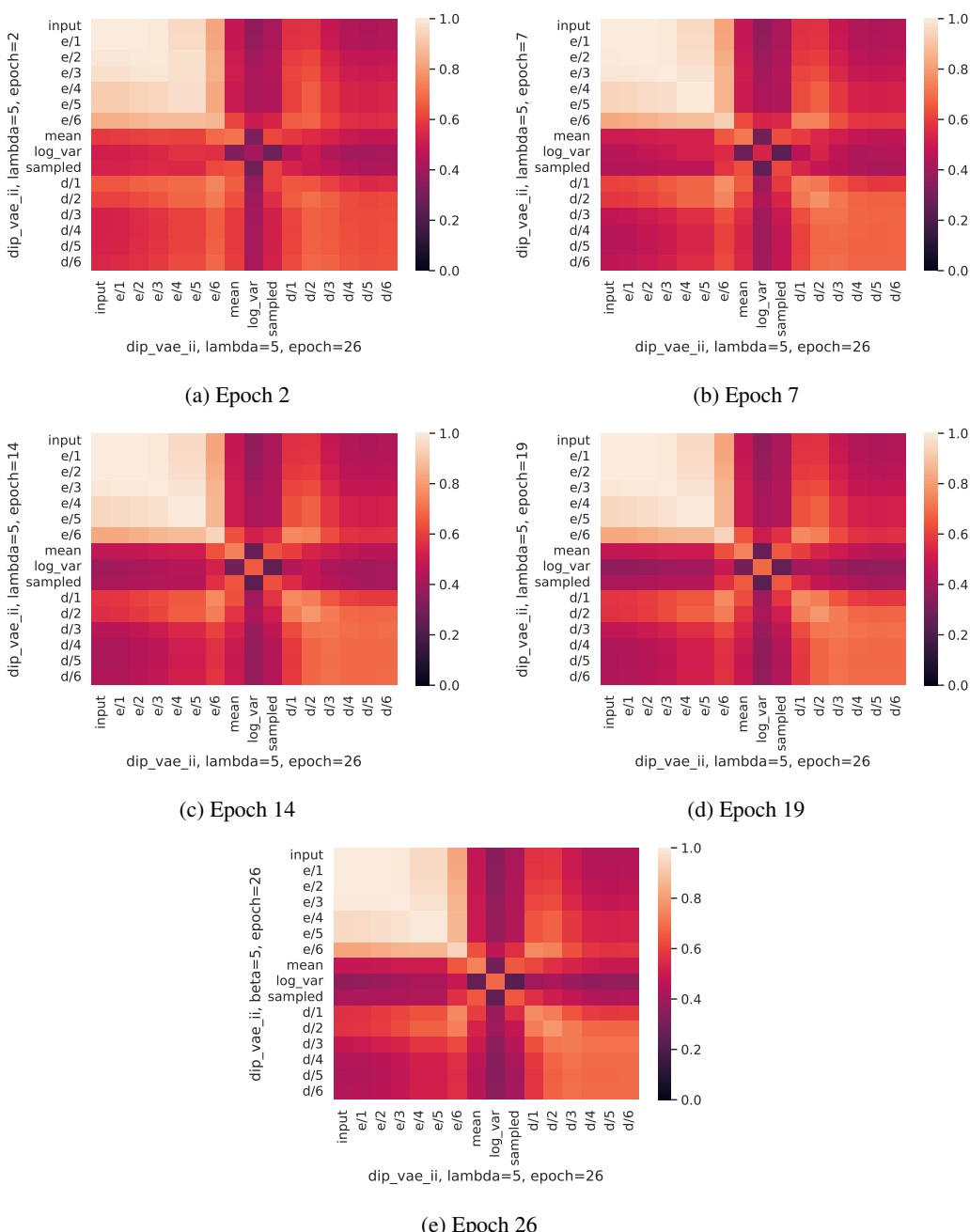

Figure 17: (a), (b), (c), (d), and (e) show the representational similarity between DIP-VAE II after full training, and at epochs 2, 7, 14, 19, and 26, respectively. All models are trained on dSprites and the results are averaged over 5 runs.

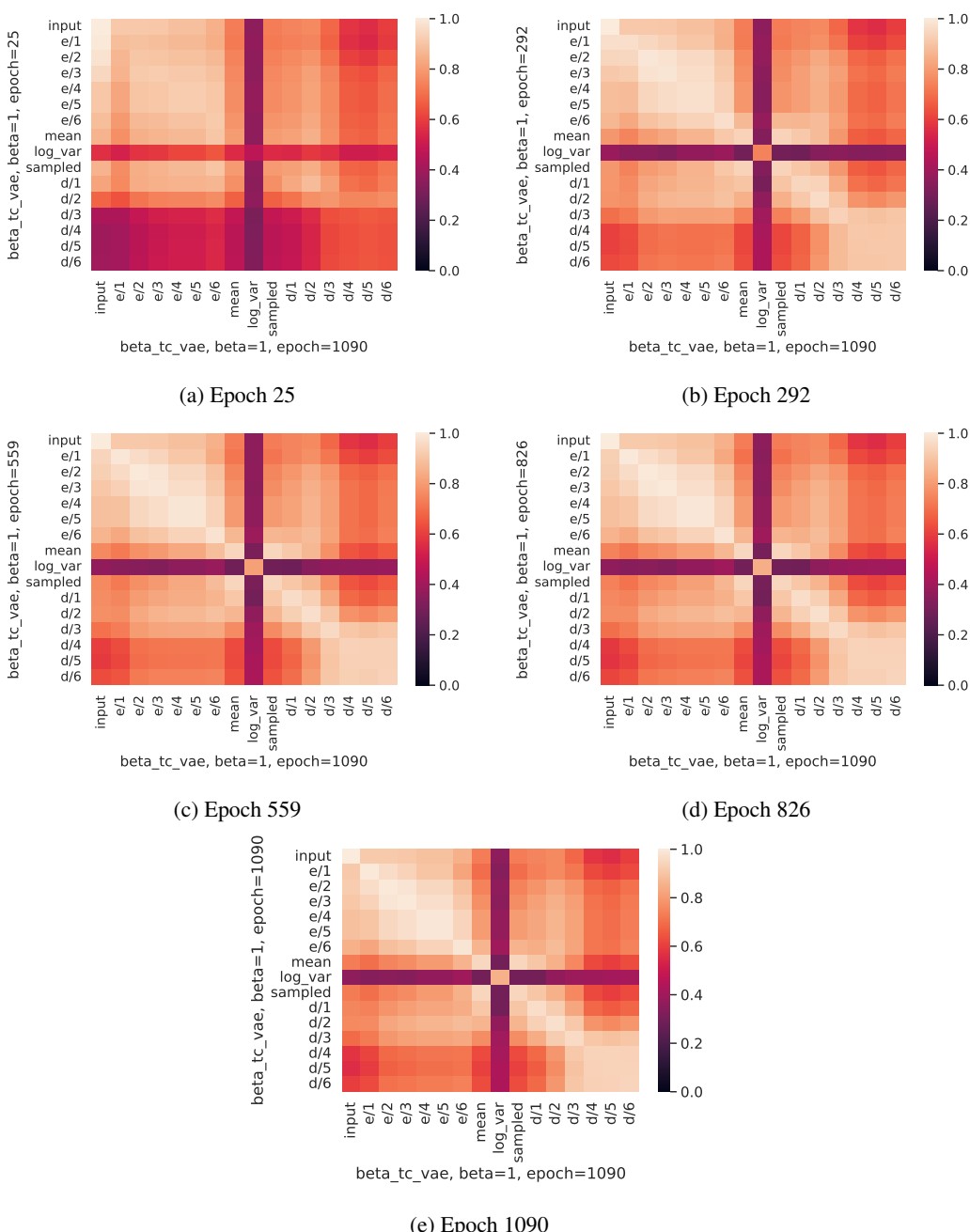

Figure 18: (a), (b), (c), (d) and (e) show the representational similarity between $\beta$-TC VAE after full training, and at epochs 25, 292, 559, 826, and 1090 respectively. All models are trained on cars3D and the results are averaged over 5 runs.

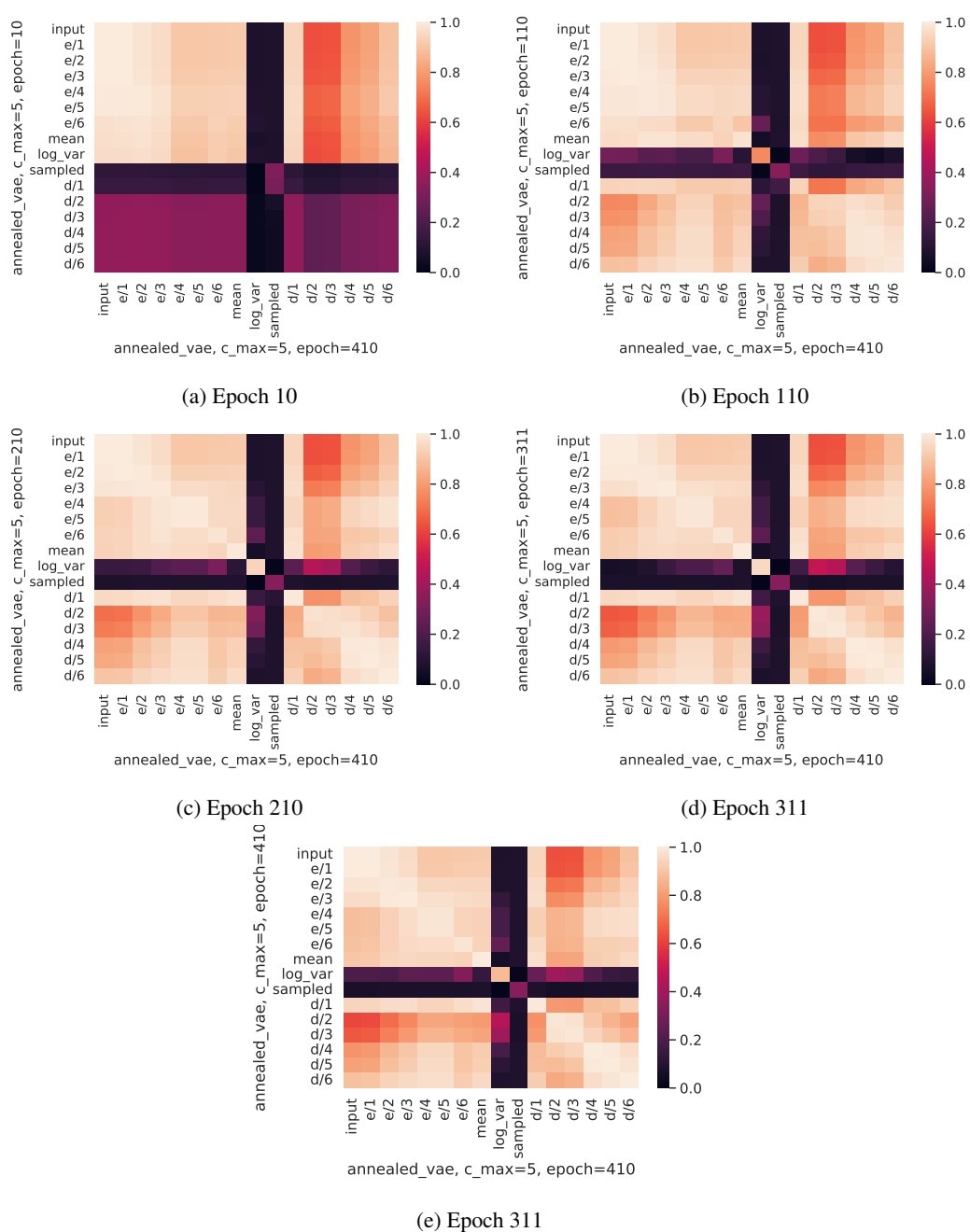

(a) Epoch 10

(b) Epoch 110

(c) Epoch 210

(d) Epoch 311

(e) Epoch 311

Figure 19: (a), (b), (c), (d) and (e) show the representational similarity between Annealed VAE after full training, and at epochs 10, 110, 210, 311, and 410 respectively. All models are trained on smallNorb and the results are averaged over 5 runs.

## J  CONVERGENCE RATE OF DIFFERENT VAES

We can see in Figure 20 that all the models converge at the same epoch, with less regularised models reaching lower losses. While annealed VAEs start converging together with the other models, they then take longer to plateau, due to the annealing process. We can see them distinctly in the upper part of Figure 20. Overall, the epochs at which the models start to converge are consistent with our choice of epoch for early training in Section 4.1.

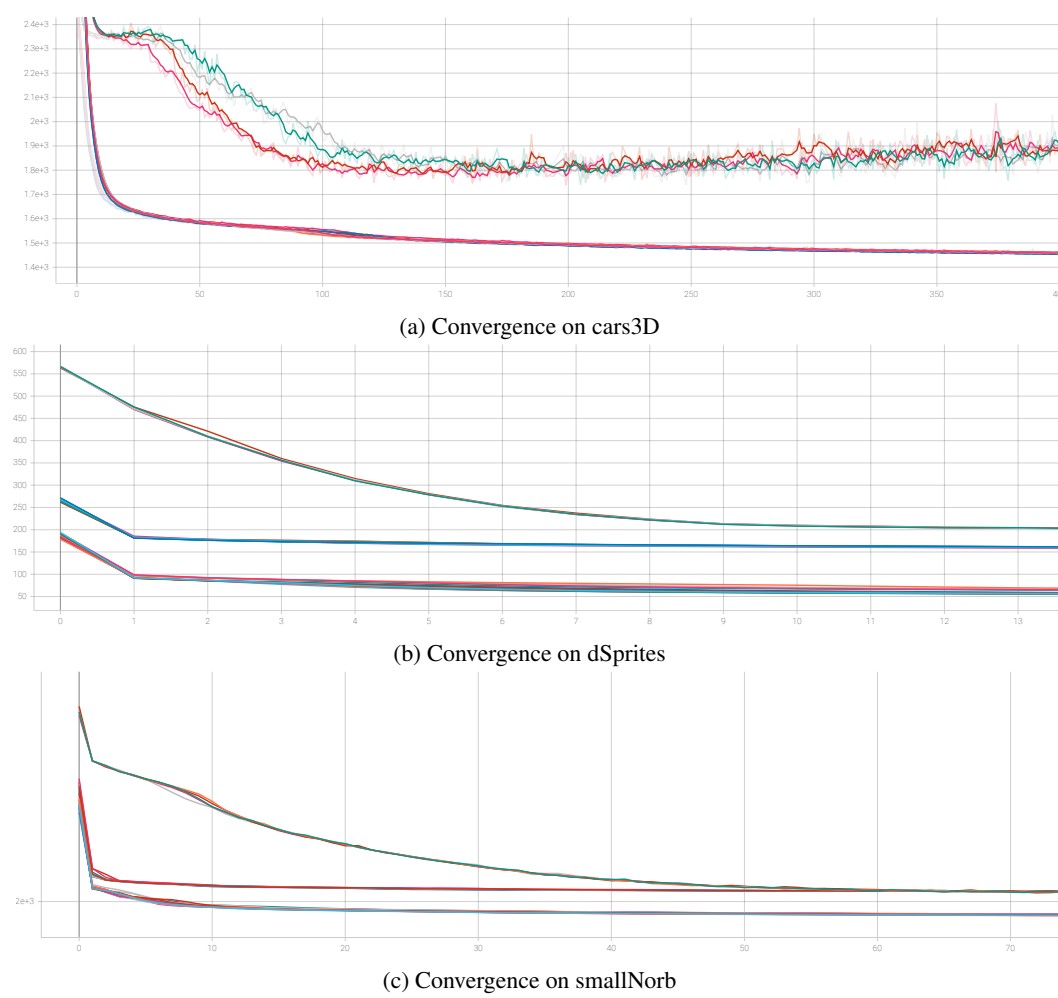

(a) Convergence on cars3D

(b) Convergence on dSprites

(c) Convergence on smallNorb

Figure 20: In (a), (b), and (c), we show the model loss of each model that converged when trained on cars3D, dSprites, and smallNorb, respectively. For each learning objective, we display 5 runs of the least and most regularised versions.

