# OpenReview forum: "How do Variational Autoencoders Learn? Insights from Representational Similarity"
_ICLR.cc/2023/Conference — Submitted to ICLR 2023_

### Official Review · Reviewer_dy86 · 2022-10-20

**Confidence:** 4
**Correctness:** 4
**Technical Novelty And Significance:** 3
**Empirical Novelty And Significance:** 4
**Recommendation:** 8

**Clarity, Quality, Novelty And Reproducibility:**

The paper is very clear, aside of the couple of points detailed above in weaknesses.

The methods are not novel in themselves, but the important thing here is that it is providing novel valuable insights into problems with VAEs and I believe this can be greatly beneficial to the community.

The paper is of good quality; methodologically sound, well written and easy to follow.

The results are highly reproducible, as the authors provide links to frameworks and results.

**Strength And Weaknesses:**

Strengths:
The paper analyses important aspects of VAEs with a principled experimwntal framework. It provides valuable and scientifically grounded explainations of problems, phenomena and trends which I personally encountered more then once in building VAEs and are, for the great part, left to intuition and practical experience to deal with. Experimentally or theoretically grounded interpretability insights are always greatly appriciated in deep learning in genereal, especially in representation/generative models.

Weaknesses:
My main concern is with the choice of experiments and the conclusions that come from it.

1) The conclusions are clear and they are indeed proven by the experiments, but either some parts of the experiments are redundant, or there is some additional conclusions that are not explained in the paper. All conclusions detailed as "implications" at the end of each experimental section only depend on comparing the same layer, e.g. e5 at 25 and 1000+ iterations, with different betas etc. So to see evidence of the claims, I only need to look at the diagonals in the matrices shown. What are the off diagonal similarities showing the reader? One interesting thing is that all encoder layers seem to get stuck to just reproduce the inputs. This is very interesting, but it is never mentioned under "implications".

2) The similarity of the representations at each layers is the main novelty aspect of the evaluation (other than the conclusions themselves), but I am not sure what motivates this choice. The more intuitive and simpler thing to do would be to compare just the similarity of the encoder/decoder weights themselves. What can we see by comparing embeddings over a dataset as opposed to weights values?

**Summary Of The Paper:**

The paper describes an evaluation framework based on representation similarity to study and explain the way VAEs learn and diagnose some of the main problems with this type of model. One aspect that sets this work apart is that it looks at representations at every layer, not just the inputs and latent space.

**Summary Of The Review:**

A good paper which peovides some useful insights into the inner working of VAEs. It is definitely a valuable contribution to ICLR and I can see researches in the area could draw from these insights to push forward the state of the art.

---

> ### Author Response · Authors · 2022-11-18
> **Answer to Reviewer dy86**
>
> Thank you for your interest in our work and for your detailed comments.
>
> Answers to Reviewer's Questions:
> ================================
>
> > All conclusions [...] showing the reader?
>
> While we cannot directly compare different types of layers, we specify whether we are interested in only the diagonal values or also off-diagonal values of the
> encoder/decoder. In the first case, we explicitly say "diagonal values", while in the second case, we mention the "top-left quadrant" for the encoder and
> "bottom-right quadrant" for the decoder. We generally do not use the top-right and bottom-left quadrants as they contain similarity
> scores between different layer architectures.
> For example, we use the off-diagonal values of the encoder representations (i.e., the similarity scores between different layers of the encoder)
> to discuss the fact that the representations learned by the encoder are similar across learning objectives (and also to classifiers with the same architectures in App. H)
> We have made this distinction clearer in the Results section.
>
> > One interesting thing [...] mentioned under "implications".
>
> Given the high similarity between the representations learned by encoders and classifiers (see App. H), it seems that
> the encoders always compress the input in the same way and the representations really change in the layer just before the
> mean representation where they vary depending on the regularisation, learning objective, etc. While practitioners often use
> pre-trained classifiers as encoders, this result shows in a principled way why such configurations are working and also suggests that
> one could freeze and reuse the same encoder for all the models trained with only last three layers (mean, variance and the layer just before)
> set as trainable layers. While this may not change much the computational time on small models, it could save computational time on more powerful
> encoder architectures.
>
> > The similarity [...] to weights values?
>
> Studying the similarity of weights is also an interesting topic, however, using similarity metrics such as Procrustes
> or CKA to monitor the weights will be challenging as we would have more features than data examples, which, as
> illustrated in Fig.1 generally results in poor estimates. Moreover, it would prevent the study of "weightless"
> elements such as sampled representations, or input images which provided important information on how the models learned.
>
> > A good [...] the state of the art.
>
> Your interest in our work is very much appreciated. We hope that our explanations and updates will alleviate your
> concerns, and we are happy to further answer any additional questions you may have.

---

### Official Review · Reviewer_2ctR · 2022-10-23

**Confidence:** 3
**Correctness:** 2
**Technical Novelty And Significance:** 2
**Empirical Novelty And Significance:** 2
**Recommendation:** 3

**Clarity, Quality, Novelty And Reproducibility:**

The paper needs improvement on explaining and discussing the relation of the results with concepts such as posterior collapse and disentanglement. The finding of the result doesn't seem to be novel -- though the method could be -- begs the question of how significant the contribution is, considering the similarity metrics are not novel either. Since code is included, I think the paper can be considered reproducible.

**Strength And Weaknesses:**

Strength:
1. detailed discussion on the limitations of the similarity metrics
2. code and libraries are provided
3. there are a diversity of datsets, similarity metrics and settings considered.

Weakness:
1. I think the similarity metric's connection with posterior collapse should be explained in greater details than references to the other papers. i don't think it's straightforward and since it is one of the main points of the paper, the motivation and justification of it could be a bit more detailed. The word "polarized regime" could also have a bit more details for the audience.
For example, the paper Figure 5 mentions "the mean and sampled representations present a growing number of passive variables which, in the case of sampled representations, leads to high dissimilarity with the input in (b)." Why does the high dissimilarity necessarily means posterior collapse, it seems like the the value is in fact converging so what if that amount of dissimilarity is just right for the model? There isn't really a threshold number that determines what a posterior collapse is for the specific problem or dataset, so i think it is ill-defined.

2. I feel that similar conclusion such as encoder learnt before decoder has been proposed or discussed or hypothesized in other literature and that should be discussed. For example, this paper "lagging inference networks and posterior collapse in variational autoencoders" is proposed based on the idea that the encoder trains slower than decoder. I think a more thorough discussion and citations in this line of work is warranted.


**Summary Of The Paper:**

The paper discussed a few similarity metrics and conducted layer-wise comparisons of VAEs using the similarity metric with different datasets, models and parameters. And conclusions are drawn on the model's disentanglement and posterior collapse based on the results. The authors found that encoder representations are learnt before decoder across all settings and they are similar across hyperparameters and learning objective within the same dataset.

**Summary Of The Review:**

Overall, I think using similarity metric could be a good idea, but the findings and their implications aren't very novel. The connection of the finding to the claims should be discussed in greater details and more rigorously too.

---

> ### Author Response · Authors · 2022-11-18
> **Answer to Reviewer 2ctR**
>
> Thank you for your interest in our work and for your detailed comments.
>
> Answers to Reviewer's Questions:
> ================================
>
> > I think the similarity metric's connection with posterior collapse [...]
>
> To clarify this, we will give more details about the polarised regime, posterior collapse, and explain how they impact CKA scores.
>
> We have seen in Sec 1. that the polarised regime can be viewed as a "healthy" version of the posterior collapse where only the latent variables that are
> not needed become passive, while posterior collapse will result in making most of the latent variables passive,
> preventing the decoder from providing a good reconstruction.
> In both cases, the passive variables are close to $\mathcal{N}(0,1)$ in the sampled representation. For this to be possible,
> the mean representation of passive variables is always very close to 0, and the variance representation to 1,
> so that the sampling process results in $z_i \approx \epsilon$ instead of $z_i = \mu_i + \sigma_i \epsilon$ where $\epsilon \sim \mathcal{N}(0,1)$.
> While passive variables exhibit differences between the mean and sampled representations, the active variables are
> very similar in mean and sampled representations. Indeed, in this case, the variance is very low, leading to $z_i \approx \mu_i$.
>
> Keeping in mind that linear CKA is a generalisation of the correlation coefficient to matrices, if there is any linear
> relationships between two representations, CKA will be high and the stronger this relationship is, the higher CKA will be.
>
> Because the mean and sampled representations have similar active variables but different passive variables, the presence of active
> variables will result in higher CKA scores. Thus, the mean and sampled representation will have higher CKA in polarised regime
> (where they contain many active variables) than during posterior collapse (where there are none or very few active variables).
> We also recommend App. G which shows that CKA can distinguish between the polarised regime and posterior collapse even when we have many
> passive variables in both cases.
>
> We have now added these details to App. G to clarify our observation.
>
> > Why does the high dissimilarity necessarily means posterior collapse [...]
>
> Given that the polarised regime is necessary for VAEs to learn properly, apart from posterior collapse,
> the only other possible scenario leading to a large number of passive variables is the case where we have more latent
> variables than needed by the VAE (in which case, some of them will be collapsed to decrease the KL divergence score)
> resulting in passive variables in mean and sampled representations, and thus different representations.
> In this case, however, the mean and sampled representations will retain some similarity with the active variables. We would thus expect a higher CKA score than in posterior collapse.
> Moreover, they will also retain a higher similarity with the input (because they still encode meaningful information).
> We show that there is indeed a clear difference of CKA scores when comparing the representations learned by $\beta$-VAEs trained on
> dSprites with 10 and 20 latent variables in appendix G of our paper, as mentioned above.
> We also observed the same trend with fully-connected VAEs in appendix F.
>
> > I feel that similar conclusion [...]
>
> We indeed believe that this is an interesting line of research and thus discussed some methods using this idea,
> such as [1], in the implication paragraph of Sec. 4.1.
> However, we are happy to extend this discussion (space permitting) if the reviewer thinks that this is not detailed enough.
>
> > The finding of the result doesn't seem to be novel [...]
>
> To the best of our knowledge, the ability of CKA to detect posterior collapse has never been discussed.
> Regarding the fact that the encoder is learned first, [1] observed lagging inference (i.e., the decoder learning before
> the encoder) for powerful decoders using a 4 dimensional toy dataset with 2 generative factors. Our results are complementary
> as they show that (1) the encoder is learned first when the decoder is not powerful enough to "bypass" this step, (2) this
> also happens for more complex datasets.
> Finally, the similarity of representations learned by VAEs with different learning objectives has, as far as we know, never
> been studied, and given that VAEs with non-conditional priors were shown to be non-identifiable [2], it is very interesting to
> see that they still have similar encoders, and, for some datasets, similar decoders. Our results imply that for some
> datasets, different learning objectives can still provide decoder representations with strong linear relationships.
>
> > Overall, [...] rigorously too.
>
> Your interest in our work is very much appreciated. We hope that our explanations and updates will alleviate your
> concerns, and we are happy to further answer any questions you may have.
>
> (references are in the next post)

---

> > ### Author Response · Authors · 2022-11-18
> > **Answer to Reviewer 2ctR (Part 2)**
> >
> >
> > References:
> > ===========
> > - [1] He, Junxian, et al. "Lagging inference networks and posterior collapse in variational autoencoders."
> > In International Conference on Learning Representations, volume 7, 2019.
> > - [2] Khemakhem, Ilyes, et al. "Variational autoencoders and nonlinear ica: A unifying framework."
> > International Conference on Artificial Intelligence and Statistics. PMLR, 2020.

---

> > ### Comment · Reviewer_2ctR · 2022-11-21
> > **My decision**
> >
> > To claim a metric can detect "posterior collapse", a relative comparison between two types of models that show slightly different values is simply not sufficient. There should be some absolute form of cut off line to claim detection. I found the contribution of the paper not rigorous or novel enough and will keep my score.

---

> > > ### Author Response · Authors · 2022-11-21
> > > **More details about CKA and posterior collapse**
> > >
> > > > To claim a metric can detect [...] cut off line to claim detection.
> > >
> > > A full posterior collapse (i.e., $z \approx \mathcal{N}(0, I)$) will lead to CKA scores close to 0 on encoder representations.
> > > This is because $z$ will be uncorrelated with any representation $y$ learned by the encoder ($z$ only depends on $\epsilon$ as shown in our previous answer) and thus,
> > > the CKA similarity scores will be 0 with any representations learned by the encoder.
> > > Given $n$ data examples, let us consider the centred representations of the latent variables, $Z \in \mathbb{R}^{n \times d}$, and of the representations learned by a given layer of the encoder, $Y \in \mathbb{R}^{n \times p}$.
> > > Now, each $Z_i \approx \mathcal{N}(0, I)$ and will be uncorrelated with every $Y_i$.
> > > Thus, as $Z$ and $Y$ are centred, we have $cov(Y,Z) = \frac{1}{n-1}Y^TZ = 0_{p \times d}$, and $cov(Z,Y) = \frac{1}{n-1}Z^TY = 0_{d \times p}$.
> > > It follows that
> > > $\parallel Y^TZ\parallel_F^2 = Tr(Y^TZZ^TY) = (n-1)^2 Tr(cov(Y,Z)cov(Z,Y)) = (n-1)^2 Tr(0_{p \times p}) = 0$.
> > > Thus, $CKA(Z, Y) = \frac{0}{||Y^TY||_F||Z^TZ||_F} = 0$, as expected, i.e., the collapsed encoder will lead to CKA of 0.

---

### Official Review · Reviewer_rD7K · 2022-10-24

**Confidence:** 3
**Correctness:** 3
**Technical Novelty And Significance:** 2
**Empirical Novelty And Significance:** 2
**Recommendation:** 5

**Clarity, Quality, Novelty And Reproducibility:**

There is not much clarity problem. And, the authors specified their experiment settings and provided a URL for the code release. Also, they promised to release the pre-trained models later on.

My concern with this paper is the quality and novelty. While the authors argue that they have tested 300 VAEs in their experiments, but the list of VAEs is pretty limited, and this limitation makes the quality, and novelty ambiguous. The authors should investigate a broader class of VAEs which contains NVAE [1] and \delta-VAE [2]. NVAE is one of the most popular VAEs these days. I wonder how the same experimental results in NVAE come out. Also, \delta-VAE is one representative model to solve the posterior collapse from the structural perspective. I wonder if the same observation can be claimed in \delta-VAE. The works [3-5] are studies to solve the posterior (or component) collapse issue in terms of utilizing other prior distributions than the Gaussian. Since the prior selection also can be considered as a hyper-parameter selection, I wonder if the claimed contribution points still hold even in the cases where other various prior distributions are assumed. What happens if the ones monitor those VAEs [2-5] with CKA score?

[1] NVAE: A Deep Hierarchical Variational Autoencoder, https://proceedings.neurips.cc/paper/2020/file/e3b21256183cf7c2c7a66be163579d37-Paper.pdf

[2] Preventing Posterior Collapse with delta-VAEs, https://openreview.net/pdf?id=BJe0Gn0cY7

[3] Stick-Breaking Variational Autoencoders, https://openreview.net/pdf?id=S1jmAotxg

[4] Dirichlet Variational Autoencoder, https://www.sciencedirect.com/science/article/pii/S0031320320303174

[5] Neural Discrete Representation Learning, https://proceedings.neurips.cc/paper/2017/file/7a98af17e63a0ac09ce2e96d03992fbc-Paper.pdf


**Strength And Weaknesses:**

The paper explores training procedures of VAE which is an interesting part, and the authors clearly list the observations that they have studied.

The authors argue that they have tested 300 VAEs, but I can not find the list of them. Does the 300(=4*5*5*3) VAEs implies the combinations on the [4 versions of VAEs of different objective functions] * [5 different initializations] * [5 different regularization hyper-parameters] * [3 datasets]?


**Summary Of The Paper:**

This paper investigates the behavior of disentangled representation learning in VAEs. Especially, the authors utilizes Centred Kernel Alignment to compare the internal behavior of VAEs. Procrustes score is mentioned in the main paper, but the authors did not utilize it due to its limitation on the computational complexity. As a result, they found that the encoder parameters are optimized earlier that the parameters in the decoder, regardless of hyper-parameters, objective function, and the datasests.


**Summary Of The Review:**

The paper investigates an interesting problem, the learning procedure of VAE. While the work shows some promising observations on the VAE learning and posterior collapse problem, the contribution is somewhat limited due to the shallow experiments. Therefore, I slightly lean to the negative side.

---

> ### Author Response · Authors · 2022-11-18
> **Answer to Reviewer rD7K**
>
> Thank you for your interest in our work and for your detailed comments.
>
> Answers to Reviewer's Questions:
> ================================
>
> > The authors argue that they have tested 300 VAEs [...]
>
> Yes, this is indeed what the 300 pre-trained models correspond to.
> We have now given the details on how this is calculated in App. E.
>
> > My concern with [...] novelty ambiguous.
>
> As explained in the computational consideration, the computational cost of this study was important and thus, we decided
> to focus on the behaviour of VAEs designed to disentangle as they possess useful properties:
> - posterior collapse situations were easy to create (with sufficiently high regularisation)
> - these models were shown to be non-identifiable (hence, not always providing the expected disentangled representations) [1], but we
> wanted to see if, despite this, they retained some similarity in the representations learned using different learning objectives.
>
> We also compared the results of the main paper with fully connected architectures in App. F and compared the representations learned
> by encoders and classifiers in App. H.
>
> > The authors should investigate [...] with CKA score?
>
> While these are interesting suggestions, we believe that the models currently used are sufficient to justify our findings. Specifically:
> - Decoders powerful enough to learn before the encoder generally result in posterior collapse if not constrained [2], and it thus
> seems to be a characteristic of well-behaved models to learn the encoder before the decoder. So,
> our observation that the encoder is learned before the decoder should hold for well-behaved models of other VAE families.
> - As posterior collapse will always result in the decoder ignoring the latent representations, we do not see any case where CKA
> would provide higher scores for collapsed representations than non-collapsed representations.
> - Given the high similarity between encoders and classifiers with equivalent architectures that was discussed in App. H, we also
> expect the encoder representations to be similar across most VAEs, though it would be interesting to see if this is also
> the case for hierarchical VAEs.
>
> Nevertheless, the study of different families of VAEs using CKA is a promising area of research.
> For example, comparison with hierarchical or semi-supervised VAEs, or VAEs using different priors could provide further
> insights into questions such as "do hierarchical VAEs learn the same representations as non-hierarchical VAEs?" or
> "Are the decoder representations learned by VAEs with different priors the same?".
> While these questions are outside this paper's scope, we have added these suggestions in the last paragraph of Sec. 5.
>
> > The paper investigates [...] the negative side.
>
> Your interest in our work is very much appreciated. We hope that our explanations and updates will alleviate your
> concerns, and we are happy to further answer any questions you may have.
>
> References:
> ===========
> - [1] Khemakhem, Ilyes, et al. "Variational autoencoders and nonlinear ica: A unifying framework."
> International Conference on Artificial Intelligence and Statistics. PMLR, 2020.
> - [2] He, Junxian, et al. "Lagging inference networks and posterior collapse in variational autoencoders."
> In International Conference on Learning Representations, volume 7, 2019.

---

### Official Review · Reviewer_nV4t · 2022-10-24

**Confidence:** 4
**Correctness:** 3
**Technical Novelty And Significance:** 3
**Empirical Novelty And Significance:** 3
**Recommendation:** 5

**Clarity, Quality, Novelty And Reproducibility:**

The paper was clearly written.  The data collect was clearly very extensive and done very professionally.  To the best of my knowledge this is the first time these measures have been used for studying VAEs.  In principle I don't see why the results would not be reproducible, although the computational cost of doing so means that I don't believe it is worth the effort.  I welcome the fact that the authors went to some length to make their raw data publicly available.

**Strength And Weaknesses:**

The paper is clearly written and presents very thorough work.  I am generally very supportive of thorough empirical studies whatever their outcome.  From the scientific point of view there is a lot to like.

My hesitation is about the usefulness of the measure of similarity.  I find the similarity measure puzzling and I feel the paper would benefit from more interpretation of what the similarity is measuring.  For example, in an auto-encoder I would naively expect that there would be similarity between the input to the encoder and the output of the decoder.  This was not displayed at all as far as I can see.  There were interpretations being made that the encoder learnt representations far faster than the decoder.  I would like to have seen some other evidence to convince me that this was not an artefact of the measure of similarity being used.

Part of the justification of the paper was as a study of disentanglement.  Indeed the paper studied a variety of VAEs that were designed to achieve disentanglement.  I found this very unconvincing.  In particular the measures were specifically designed to be invariant to  rotations in the representations as I understand it.  However rotations in latent space are critical to disentanglement so it seemed like a strange measure to investigate disentanglement.  Apart from a discussion in the introductions I could not see that the paper threw any light on disentanglement.

**Summary Of The Paper:**

The paper covers a very extensive empirical study of the representational similarity among different layers between auto-encoders at different stages of training and between different auto-encoders.  The study involves two different measures of similarity.

**Summary Of The Review:**

Although there is much to be admired about the paper, I still need to be convinced about its utility.  It describes two related measure of similarity, however, interpreting what these measures tell us is very difficult.  Without a better explanation of what we are meant to take away from these measures I struggle to gain much insight.  I would have preferred to see a more holistic analysis that could be much more limited, but would provide more insight into how we should interpret these similarity measures and what they tell us.  It would have been helpful to see the similarity plot of a model with itself at the same point in time as this would provide a baseline to understand the results shown.  There seems to be a sudden loss of similarity as the representation hits the decoder.  Given that auto-encoders preserve a lot information it would have been really useful to understand what is going on.  Finally, the paper is built up as an exploration of disentanglement, but it is not at all obvious that the paper has anything to say about this.

Given the huge work that has gone into this, I would like to be persuaded to increase my score.  But to do this I need more help in understanding the utility of the work that has been carried out.

---

> ### Author Response · Authors · 2022-11-18
> **Answer to Reviewer nV4t**
>
> Thank you for your interest in our work and for your detailed comments.
>
>
> Answers to Reviewer's Questions:
> ================================
> > My hesitation is about [...] what the similarity is measuring
>
> Linear CKA is a generalisation of the correlation coefficient to matrices. If there is any linear
> relationship between two representations, CKA will be high and the stronger this relationship is, the higher CKA will be.
>
> We have added this explanation to the background section to provide a more intuitive view of what CKA does.
>
> > I would [...] similarity between the input to the encoder and the output of the decoder.
>
> This can be observed in Fig.3., for example, by looking at the similarity between the output of the decoder (d/6) and the input.
> One can see that this is much lower at early epochs (top-right cell) than at the end of the training (bottom-left cell),
> showing that the decoder learned to better reconstruct the input after the early epochs.
>
> > There were interpretations [...] artefact of the measure of similarity being used.
>
> To demonstrate this, we recommend App. I, especially Fig. 17, 18 and 19 which clearly show the evolution
> of the decoder over multiple epochs. In the top left quadrant, we can see that the similarity between epochs of the representations learned by the encoder becomes high quickly (i.e., the representations do not change much between epochs).
> On the bottom right quadrant however, the decoder clearly continues to learn for a larger number of epochs,
> with increasing similarity scores as the training goes on.
>
> > Part of the justification [...] a strange measure to investigate disentanglement
>
> Our goal in this paper was not to measure disentanglement. Indeed, CKA would not distinguish between disentangled and entangled
> representations, as explained in 2.3. Instead, we wanted to study the behaviour of VAEs designed to disentangle as they possess useful properties:
> - posterior collapse situations were easy to create (with sufficiently high regularisation)
> - these models were shown to be non-identifiable (hence, not always providing the expected disentangled representations) [1], but we
> wanted to see if, despite this, they retained some similarity in the representations learned using different learning objectives.
>
> We believe that the high similarity observed between the representations learned by VAEs with different learning objectives
> is insightful in that regard. Indeed, our results imply that on some datasets, different learning objectives can still
> provide decoder representations with strong linear relationships.
>
> We have now made this clear in the background section.
>
> > It would have been helpful [...] to understand what is going on.
>
> While we could not add the similarities of models with themselves due to space limitations, we have added some heatmaps
> of models compared to themselves in App. I. to clarify this.
>
> > Given the [...] carried out
>
> Your interest in our work is very much appreciated. We hope that our explanations and updates will alleviate your
> concerns. We are happy to further answer any questions you may have.
>
>
> References:
> ===========
> [1] Khemakhem, Ilyes, et al. "Variational autoencoders and nonlinear ica: A unifying framework."
> International Conference on Artificial Intelligence and Statistics. PMLR, 2020.

---

> > ### Comment · Reviewer_nV4t · 2022-11-21
> > **Response**
> >
> > Thank you for trying to address my concerns.  I have to say I still find the measures you are using confusing.  I understand that your measurements are picking up linear relationships, but I'm still at a loss as to how to interpret that intuitively.  As I said it is puzzling that there is a low score between the output of the decoder and the input into the encoder.  Although there may be a few instances where the score is not that low, in many cases the scores increase as you go through the decoder (e.g. Figure 3(a)).  This is strange behaviour for an autoencoder.  Maybe because the only non-linearities come from ReLUs and pooling/striding then your measurements are proxies of the useful information being stored in each layer.  This might progressively decrease through the network.  However, from the results you give it is really difficult to know what is happening.  Your justification for studying different auto-encoders for doing disentanglement still puzzles me.  It does not help that the first line of you abstract talks about this.  I am sorry, but a remain as confused as before.  You state in the abstract the the behaviour of VAEs doing disentanglement is not yet fully understood.  I don't think you have made a convincing case that they are any better understood after this paper.  Instead you have provided a set of rather baffling results showing that the measure you use has different behaviour on different datasets and for different VAEs.  You might be able to interpret these many graphs is some profound way that gives great insight, but despite my best efforts I am still struggling to make any sense of them.

---

> > > ### Author Response · Authors · 2022-12-08
> > > **Additional clarifications**
> > >
> > > > Thank you for trying [...] as you go through the decoder (e.g. Figure 3(a)).
> > >
> > > We are not sure what the reviewer mean by "scores increase as you go through the decoder (e.g. Figure 3(a))" ? Could they
> > > explain which part of Fig. 3(a) they are referring to ?
> > > In Fig. 3(a), we compare the representations learned by the same VAE after 25 and 1090 epochs (x and y axis, respectively).
> > > The CKA score between the input and decoder output at epoch 25 (top-right point x=d/6, y=input) is around 0.2 while
> > > the score at epoch 1090 (bottom-left point at x=input, y=d/6) is close to 0.6. Thus, the decoder improves its reconstruction
> > > as the training goes on. We can also see when comparing the similarity scores of the encoder representations
> > > (top-left quadrant, x and y values starting with e/) that they have a high similarity ($>0.8$) between epoch 25 and 1090,
> > > indicating that they did not change much after epoch 25.
> > > On the other hand, the similarity between decoder representations (bottom-right quadrant, x and y values starting with d/)
> > > is lower ($\approx 0.6$) indicating that the decoder representations changed between epoch 25 and 1090.
> > > Given this, we can conclude that the encoder is learned before the decoder.
> > >
> > > > Your justification [...] yet fully understood.
> > >
> > > We discuss disentanglement capacity of VAEs in the first line of our abstract as one of the thing that makes
> > > them popular, not as the topic of our paper. When we said "However, their behaviour is not yet fully understood",
> > > we meant the behaviours of VAEs not specifically their ability to disentangle.
> > > However, we are happy to reformulate this to avoid any confusion.
> > >
> > > > I don't [...] after this paper.
> > >
> > > Our objective is not to shed light into disentangled representation learning but to have a better understanding of
> > > the learning process of VAEs in general, which we chose to investigate with representational similarity techniques.
> > >
> > > > Instead you have provided a [...] for different VAEs.
> > >
> > > We would like to emphasise that CKA and Procrustes results were consistent in our experiment
> > > (see App. D) indicating that our observations were not an artifact of CKA. We generally observed the same trends across
> > > learning objectives and datasets except for the representational similarity of the mean, variance, sampled and decoder
> > > representations across learning objectives which was different depending on the dataset, indicating that different
> > > learning objectives may find different local optima for a given dataset, consistently with previous conjectures [1].
> > >
> > > > You might be able [...] any sense of them.
> > >
> > > We are sorry that our previous answers did not clarify the questions you had, and we hope that our current answer will
> > > be more helpful. We are happy to further discuss any points that remain unclear.
> > >
> > >
> > > References
> > > ===========
> > > - [1] Dominik Zietlow, Michal Rolinek, and Georg Martius. Demystifying Inductive Biases for (Beta-)
> > > VAE Based Architectures. In Proceedings of the 38th International Conference on Machine
> > > Learning, volume 139 of Proceedings of Machine Learning Research, 18–24 Jul 2021.
> > > - [2] Alexander A Alemi, Ian Fischer, Joshua V Dillon, and Kevin Murphy. Deep Variational Information
> > > Bottleneck. In International Conference on Learning Representations, volume 5, 2017.

---

### Official Review · Reviewer_hNQM · 2022-10-25

**Confidence:** 4
**Correctness:** 3
**Technical Novelty And Significance:** 1
**Empirical Novelty And Significance:** 2
**Recommendation:** 5

**Clarity, Quality, Novelty And Reproducibility:**

Clarity:
The paper is well-written. The advantages and limitations of the proposed metrics are clearly discussed and the experimental results are discussed in-line with the limitations. The reason to choose specific metrics can be elaborated.

Quality:
The experimental set-up is outlined clearly. The experiments are detailed. The approach and the metrics are well-explained.

Novelty:
CKA has been used in  [1] to study the representations of different deep learning frameworks (not VAE in particular).
The paper does not provide a new streamlined procedure or any new insights over the former. Therefore, while the findings are well demonstrated through experiments, the originality is limited.
[1] Similarity of Neural Network Representations Revisited. ICML 2019.


Reproducibility:
The work should be reproducible.


**Strength And Weaknesses:**

Strengths:

+The work is well-written.
+The limitations of the approach are clearly discussed. The analysis are performed taking into account the limitations of the proposed metrics.
+The results seem like they should be reproducible given sufficient compute.

Weaknesses:
-The approach is limited to the comparison of representations from similar layers. For example, the similarity between convolutional and deconvolutional layers cannot be evaluated using the proposed approach.
- The findings in the paper, regarding posterior collapse, and encoder learning before decoder to prevent posterior collapse are already well-established in the community and it is not clear what is the main "new" finding of the paper. The metrics, especially CKA [1] have been used to study deep networks in prior work and the work applies the same tools to VAEs, thus there are no new contributions.
 - CKA is claimed to be a metric of choice for monitoring posterior collapse. What makes it a better tool? Why can we just monitor the latents?
- The hyperparameters considered for evaluation is limited to choice of regularization? There are other factors such as optimizers, lr schedulers etc which also can impact training. What are the insights here?

[1] Similarity of Neural Network Representations Revisited. ICML 2019.


**Summary Of The Paper:**

This work studies the layer-wise representations of different VAEs. The similarity between the layer-wise representations is measured using the Centered Kernel Alignment metric and the Procrustes scores. Experiments are performed to study the effect of different hyperparameters on the same model, regularization, initialization, and learning objectives.


**Summary Of The Review:**

Overall, the paper is well-written and the experiments are conducted to evaluate the similarity of different representations from the VAEs. The findings in the paper such as  "encoders learn before decoders" are already well-known in the community. Moreover, the paper argues that linear CKA can be used to track posterior collapse. The motivation for this claim is not clear. Experiments are limited to certain design choices, why only choose regularization as a hyperparameter?

---

> ### Author Response · Authors · 2022-11-18
> **Answer to Reviewer hNQM**
>
> Thank you for your interest in our work and for your detailed comments.
>
> Answers to Reviewer's Questions:
> ================================
> >  The findings [...] main "new" finding of the paper.
>
> To the best of our knowledge, the ability of CKA to detect posterior collapse has never been discussed.
> Regarding the fact that the encoder is learned first, [1] observed lagging inference (i.e., the decoder learning before
> the encoder) in powerful decoders using a 4 dimensional toy dataset with 2 generative factors. Our results are complementary
> as they show that (1) the encoder is learned first when the decoder is not powerful enough to "bypass" this step, (2) this
> also happens in more complex datasets.
> Finally, the similarity of representations learned by VAEs with different learning objectives has, as far as we know, never
> been studied, and given that VAEs with non-conditional priors were shown to be non-identifiable [2], it is very interesting to
> see that they still have similar encoders, and, on some datasets, similar decoders. Our results imply that on some
> datasets, different learning objectives can still provide decoder representations with strong linear relationships.
>
> > The metrics [...] no new contributions.
>
> Our objective was not to extend CKA but to use it to study VAEs using CKA, which, to the best of our knowledge, has not been done before.
> As discussed above, our observations provide complementary insights into well-known phenomenon, a principled way to explain behaviours that are often encountered empirically (e.g., encoder representations are similar to each others and to classifiers),
> and new results on the similarity of the decoders across datasets which could be interesting for the research community working on
> identifiable representation learning.
>
> > CKA is claimed [...] better tool?
>
> While CKA can be used to monitor posterior collapse, we made no claim about it being a better way of doing it.
> We think it could be complementary to metrics that are more computationally expensive such as Mutual Information (MI) when the
> computational time is a constraint (e.g., if this is used during model training).
> However, MI would detect non-linear relationships while CKA cannot, which makes it more useful in contexts where the computational
> time is not an issue.
>
> > Why can't we just monitor the latents?
>
> One could indeed monitor the latents (e.g., checking the mean and variance representations of the data examples and count the number
> of dimensions for which they are consistently close to 0 and 1, respectively). However, it is quite hard to distinguish between a case
> of posterior collapse and a case of polarised regime where the number of latent dimensions is set to a large value using
> this technique. App. G, shows that CKA can successfully distinguish between these two cases.
>
> > The hyperparameters considered [...] insights here?
>
> As explained in the computational considerations, the computational cost of this study was important and thus, we decided
> to focus on the behaviour of VAEs designed to disentangle as they possess useful properties:
> - posterior collapse situations were easy to create (with sufficiently high regularisation)
> - these models were shown to be non-identifiable (hence, not always providing the expected disentangled representations) [2], but we
> wanted to see if, despite this, they retained some similarity in the representations learned using different learning objectives.
>
> While we did not test lr schedulers or optimisers changes, we also compared the results of the main paper
> with fully connected architectures in App. F and compared the representations learned by encoders and classifiers in App. H.
>
> > Overall, [...] a hyperparameter?
>
> Your interest in our work is very much appreciated. We hope that our explanations and updates will alleviate your
> concerns, and we are happy to further answer any questions you may have.
>
> References:
> ===========
> - [1] He, Junxian, et al. "Lagging inference networks and posterior collapse in variational autoencoders."
> In International Conference on Learning Representations, volume 7, 2019.
> - [2] Khemakhem, Ilyes, et al. "Variational autoencoders and nonlinear ica: A unifying framework."
> International Conference on Artificial Intelligence and Statistics. PMLR, 2020.

---

### Author Response · Authors · 2022-11-17
**Summary of Revision 1**

Dear reviewers,
Thank you for your time and effort.
We have implemented your suggestions, and we uploaded a revised version of our paper to openreview along with a diff file in [supplementary material](https://openreview.net/attachment?id=s_2Rye-RctO&name=supplementary_material) to provide a clear overview of all our changes.

Here are some short answers to questions and comments that appeared in multiple reviews:
- **How does CKA work**: Linear CKA is a generalisation of the correlation coefficient to matrices. So, if there is any linear
relationship between two representations, CKA will be high and the stronger this relationship is, the higher CKA will be (like Pearson's correlation measures a linear relationship between two scalar variables).
- **Novelty of contributions**: As detailed in individual answers, our observations provide complementary insights into well-known phenomenon, a principled way to explain certain
behaviour of VAEs that are often encountered empirically (e.g., encoder representations are similar to each other and to classifiers), and new results on the similarity of the decoders across datasets, which could be useful for the research community working on identifiable representation learning.
- **Using CKA for posterior collapse**: In Appendix G, we show that CKA can distinguish between the polarised regime and posterior collapse and have now further explained how CKA can efficiently detect collapsed models.

More details on other changes will be made available on the 18th of November in dedicated answers to individual reviewers.
Meanwhile, the updates can be seen in the diff file provided in [supplementary material](https://openreview.net/attachment?id=s_2Rye-RctO&name=supplementary_material).
We hope that our answers and updates will clarify the paper and alleviate any concerns you had.
We are available to answer any other questions you may have.

---

### Decision · Program_Chairs · 2023-01-20

**Decision:**

Reject

**Justification For Why Not Higher Score:**

There were two consistent concerns among reviewers.

**Justification For Why Not Lower Score:**

N/A

**Metareview: Summary, Strengths And Weaknesses:**

This paper studies layer-wise representation in VAEs using the Centered Kernel Alignment metric and the Procrustes scores. While there were some strengths to the paper, there was two consistent concerns among the reviewers: novelty and significance. For the former, it was mentioned that the metrics and methods do not appear to be model and that some of the findings in the paper such as "encoders learn before decoders" are already well-known in the community. For the latter, it seemed unclear what utility the proposed method brings to users. I hence recommend rejecting this paper.